# Rhodopsin-cyclases for photocontrol of cGMP/cAMP and 2.3 Å structure of the adenylyl cyclase domain

Ulrike Scheib[1], Matthias Broser[1], Oana M. Constantin[2], Shang Yang[3], Shiqiang Gao[3] Shatanik Mukherjee[1], Katja Stehfest[1], Georg Nagel[3], Christine E. Gee[2] & Peter Hegemann[1]

The cyclic nucleotides cAMP and cGMP are important second messengers that orchestrate fundamental cellular responses. Here, we present the characterization of the rhodopsin-guanylyl cyclase from *Catenaria anguillulae* (CaRhGC), which produces cGMP in response to green light with a light to dark activity ratio >1000. After light excitation the putative signaling state forms with $\tau = 31$ ms and decays with $\tau = 570$ ms. Mutations (up to 6) within the nucleotide binding site generate rhodopsin-adenylyl cyclases (CaRhACs) of which the double mutated YFP-CaRhAC (E497K/C566D) is the most suitable for rapid cAMP production in neurons. Furthermore, the crystal structure of the ligand-bound AC domain (2.25 Å) reveals detailed information about the nucleotide binding mode within this recently discovered class of enzyme rhodopsin. Both YFP-CaRhGC and YFP-CaRhAC are favorable optogenetic tools for non-invasive, cell-selective, and spatio-temporally precise modulation of cAMP/cGMP with light.

[1] Institute for Biology, Experimental Biophysics, Humboldt-Universität zu Berlin, 10115 Berlin, Germany. [2] Institute for Synaptic Physiology, Center for Molecular Neurobiology Hamburg, University Medical Center Hamburg-Eppendorf, 20251 Hamburg, Germany. [3] Department of Biology, Institute for Molecular Plant Physiology and Biophysics, Biocenter, Julius-Maximilians-University of Würzburg, Julius-von-Sachs-Platz 2, 97082 Würzburg, Germany. These authors contributed equally: Christine E. Gee, Peter Hegemann. Correspondence and requests for materials should be addressed to C.E.G. (email: christine.gee@zmnh.uni-hamburg.de) or to P.H. (email: hegemann@rz.hu-berlin.de)

The cyclic nucleotides cAMP and cGMP are ubiquitous second messengers that regulate essential cellular processes including basic metabolism, gene expression, differentiation, proliferation, and cell survival[1–4]. Spatial and temporal segregation of cyclic nucleotides and their effectors allows precise coordination of a multitude of signaling pathways[5]. Despite the enormous impact of cyclic nucleotides many questions about their exact roles remain unanswered. Pharmacological approaches (e.g., forskolin, IBMX) do not allow cell specific manipulation of cyclic nucleotides in tissue and lack precision in space and time, limitations that can be overcome using light-activated enzymes.

The advent of optogenetic tools has allowed precise spatio-temporal control of cellular processes. For example, non-invasive control of intracellular [cAMP] is achieved using the soluble photoactivatable adenylyl cyclases from *Euglena* (euPAC) or *Beggiatoa* (bPAC)[6–10]. However, the cytoplasmic localization, and slow off-kinetics of these flavin-based enzymes (bPAC $\tau_{off}$ = 12 s) are disadvantageous characteristics to study fast cyclic nucleotide signaling near cellular membranes. Recently, BeRhGC (a.k.a. BeGC1, CyclOp, or RhoGC), a rhodopsin directly connected via a linker to a guanylyl cyclase domain, was discovered in the aquatic fungus *Blastocladiella emersonii*[11]. BeRhGC converts GTP into cGMP in many cell types upon green light stimulation whereas it is totally inactive in the dark[12–14]. Upon excitation the putative rhodopsin signaling state is formed within 8 ms and decays after 100 ms, whereas the enzyme's activity declines with a $\tau$ of ~300 ms. BeRhGC is however, only moderately resistant to high light intensities and bleaches in continuous light[13,15].

In this study, we further characterize the RhGC from the fungus *Catenaria anguillulae* (CaRhGC), which is another member of the chitin-walled *Blastocladiomycota*[11,12], and the adenylyl cyclase CaRhAC resulting from the point mutations E497K, C566D in *Xenopus* oocytes and hippocampal neurons. We additionally present the crystal structure of the ligand-bound adenylyl cyclase domain (CaAC) at 2.25 Å resolution, which reveals the mechanistic basis for the change from cGMP to cAMP production. The rhodopsin domain from *Catenaria* is more photostable than that from *Blastocladiella*, and the signaling state persists longer, both of which might be highly desirable traits for optogenetic applications. YFP-CaRhGC together with the YFP-CaRhAC mutant, expand the optogenetic toolbox allowing control of cAMP/cGMP signaling within milliseconds close to cellular membranes.

## Results

**Characterization of CaRhGC in oocytes and rat neurons.** Comparing the amino acid sequences of CaRhGC and BeRhGC (a.k.a. CaCyclOp and BeCyclOp, or BeGC1 and CaGC1) revealed 77% identity (Supplementary Fig. 1). Most differences occur within the N-terminal extension (i.e., amino acids 72–170), which is predicted to harbor 1–2 extra helices upstream of the 7 transmembrane helices that characterize the rhodopsins (Fig. 1a, Supplementary Figs. 1 and 2[12,13]). A second region of variation occurs in helices 4 and 5 including the helix 4/5-loop. Additionally, we found a higher probability for coiled-coil formation of the N-terminal helix-1 of CaRhGC compared to BeRhGC (Supplementary Tables 1 and 2). To localize the N terminus, we inserted the rhodopsin domain of CaRhGC (aa 1–425) between the two fragments of a split YFP. The observed YFP fluorescence confirmed that the extra N-terminal segment spans the membrane in such a way that the N terminus is positioned intracellularly (Fig. 1b).

We co-expressed full-length CaRhGC and the cGMP-sensitive cyclic nucleotide-gated A2 channel from rat olfactory neurons (CNG(cGMP), $K_{1/2}^{cAMP}$ = 36 μM, $K_{1/2}^{cGMP}$ = 1.3 μM)[16] in

*Xenopus* oocytes and recorded inward photocurrents in response to 2 second green light pulses (Fig. 1c). The photocurrents were light intensity-dependent with half maximal saturation for the slope (EC$_{50}$) at 0.027 mW mm$^{-2}$ (Fig. 1d) and declined after light-off with an apparent $\tau_{off}$ = 9.2 ± 1.7 s ($n$ = 8, 0.12 mW mm$^{-2}$). CaRhGC is highly selective for GTP and no photocurrents were recorded from oocytes co-expressing CaRhGC with the cAMP-sensitive CNGA2 channel (C460W/E583M, CNG(cAMP), $K_{1/2}^{cAMP}$ = 0.89 μM, $K_{1/2}^{cGMP}$ = 6.2 μM)[16] (Fig. 1c), which also excludes that CaRhGC itself is conductive.

To quantify cyclic nucleotide concentrations, cGMP and cAMP were determined in oocyte lysates using an enzyme-linked immunosorbent assay (ELISA) (Fig. 1e, f). In CaRhGC-expressing cells the dark concentration of cGMP remained unchanged at 0.4 pmol/oocyte, whereas illumination for 1 min with green light increased cGMP to 120 pmol/oocyte. To facilitate future studies in host cells, including neuronal networks, we labeled CaRhGC N-terminally with YFP, which did not change enzyme activity. In contrast, C-terminal YFP tagging caused the enzyme to be partially active in the dark increasing cGMP concentration 10 fold (Fig. 1e). For all CaRhGC variants, cAMP concentrations were unaffected (Fig. 1f), confirming the GTP selectivity of CaRhGC[12].

Hippocampal neurons expressing YFP-CaRhGC, CNG(cGMP) and mtSapphire had normal morphology and normal whole-cell voltage responses to current injection (Fig. 2a–c). Flashes of green light repeatedly evoked transient currents through the cGMP-sensitive channels (Fig. 2c). The photocurrents evoked by strong green light in YFP-CaRhGC-expressing neurons were similar in size to the photocurrents recorded from neurons expressing BeRhGC, (Fig. 2d, e; Supplementary Table 3, Supplementary Fig. 3a, b). Interestingly, without the YFP-tag photocurrents were of similar amplitude but evoked in only 30% of neurons expressing CaRhGC (Fig. 2d, e; Supplementary Table 3). We attempted to add a C-terminal mycHis tag to YFP-CaRhGC and this also had a negative impact on reliability (Supplementary Table 3). But whether codons were optimized for human or mouse expression was unimportant (Supplementary Table 3).

Interestingly, the kinetics of the guanylyl cyclases from *Blastocladiella* and *Catenaria* were quite different (Fig. 2f, Supplementary Fig. 3c, d). The time to photocurrent onset was significantly shorter in neurons expressing CaRhGC than in neurons expressing BeRhGC (median time to onset CaRhGC 23 ms, BeRhGC 120 ms, $p$ = 0.0029, Kruskall–Wallis). In addition, the slope was greater indicating that the cGMP concentration increases faster in neurons expressing the *Catenaria* guanylyl cyclase than it does in neurons expressing the cyclase from *Blastocladiella* (Fig. 2f). Moreover, the CaRhGC photocurrent decay was much faster in neurons than in oocytes ($t_{1/2}$ ~0.2 s, Supplementary Fig. 3d), suggesting that phosphodiesterases rapidly degrade cGMP in neurons. The YFP-CaRhGC photocurrent amplitudes and slopes were graded with the light intensity with an EC$_{50}$ of 0.7 mW mm$^{-2}$ (Fig. 2g, h). No photocurrents were evoked in neurons transfected with CaRhGC or YFP-CaRhGC and CNG(cAMP), confirming that the specificity for producing cGMP is unchanged in neurons (Fig. 2d, e; Supplementary Table 3).

**Spectroscopic properties of the *Ca* rhodopsin domain.** To assess the spectral properties of CaRhGC, we purified the recombinant rhodopsin fragment CaRh (amino acid residues 1 to 396) from insect cells (Sf9). Dark-adapted CaRh showed a typical unstructured rhodopsin spectrum with a maximum at 540 nm (D$_{540}$, Fig. 3a). Bright green light (530 nm) converted D$_{540}$ into a light-adapted species with slightly shifted absorption maximum

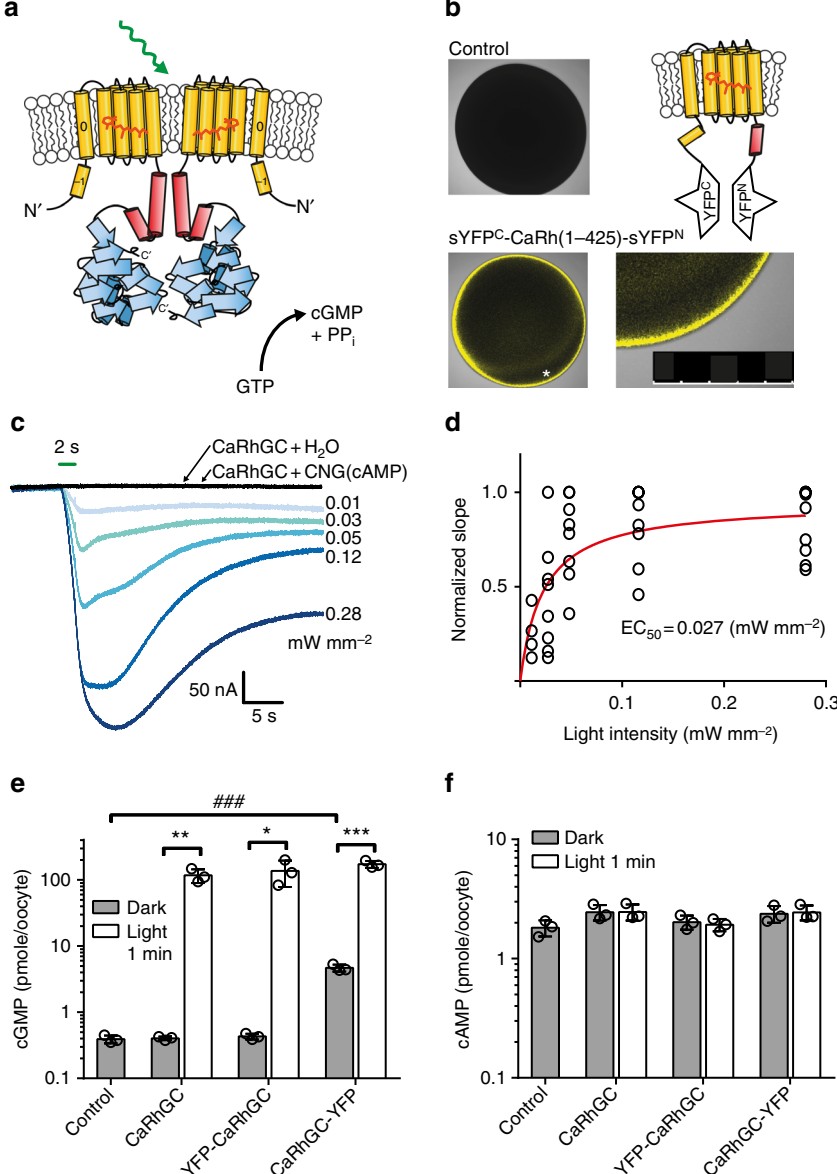

**Fig. 1** Activity of CaRhGC in *Xenopus* oocytes. **a** Model of the dimeric rhodopsin-guanylyl cyclases from *Catenaria anguillulae* (Ca). The photo-sensitive rhodopsin domain (yellow) is directly connected to the guanylyl cyclase domain (blue) via a coiled-coil stretch (red). **b** Single plane confocal images of a non-injected oocyte (top left) and an oocyte expressing the first 1–425 aa (CaRh) inserted between the two halves of a split YFP (top right, bottom left, bottom right: *region magnified, scale bar = 500 μm). YFP fluorescence is reconstituted in oocytes 3 days after injection. **c** Representative currents from a *Xenopus* oocyte, expressing CaRhGC together with the cGMP-sensitive CNGA2 channel in response to green light (blue traces, light 2 s, 560 ± 60 nm, intensities as depicted). No photocurrents were detected in oocytes expressing CaRhGC alone or together with a cAMP-sensitive CNGA2 channel (gray traces, 2 s, 560 ± 60 nm, 0.28 mW mm$^{-2}$). **d** Half saturation of current initial slopes (EC$_{50}$), deduced from **c** was reached at 0.027 mW mm$^{-2}$, the light intensity-response relationship was fitted exponentially. **e**, **f** ELISA-based quantification of cGMP (**e**) and cAMP (**f**) from whole oocyte lysates. Oocytes expressing untagged CaRhGC, YFP-tagged CaRhGC (N- or C-terminal) were kept in darkness or illuminated with green light (1 min, 532 nm 0.3 mW mm$^{-2}$). Data are presented as mean ± s.d., $n = 3$ samples of 5 oocytes each, ***$p = 0.0002$, **$p = 0.002$, *$p = 0.02$, unpaired *t*-tests light vs dark. ###$p = 0.0001$, Dunnett's multiple comparisons vs control (dark conditions, following one-way ANOVA ($p < 0.0001$)). Control = non-injected oocytes, CNG = channel cyclic nucleotide-gated channel

(L$_{538}$) (inset Fig. 3a), but caused very little bleaching even after long exposure[17]. To characterize CaRh photocycle intermediates, absorption changes were recorded from 100 ns to 10 s after stimulation with 10 ns 530 nm laser flashes (Fig. 3b, c). Evolution-associated difference spectra (EADS) of the intermediates and their life times (τ values) were extracted using a global fit routine (Fig. 3b). We first detected an early red-shifted K-like photoproduct K$_{600}$, that rose faster than our time resolution and decayed at pH 7.5 with τ = 0.81 μs into a blue-shifted

intermediate (L1$_{450}$), which was not observed in the BeRh photocycle[13]. EADS and the extracted time trace at 458 nm (Fig. 3b) revealed accumulation of a second L2$_{450}$ state with τ = 397 μs, which converted into the proposed signaling state M$_{380}$ with τ = 31 ms (compared to 8 ms for BeRhGC[13]). The temporal dependence of the absorbance changes at key wavelengths (Fig. 3c), which refer to the photocycle intermediates, revealed a slower decay of the M$_{380}$ state (τ = 571 ms) compared to BeRh (τ = 100 ms)[13]. In summary, green light converts CaRh D$_{540}$ to a red-

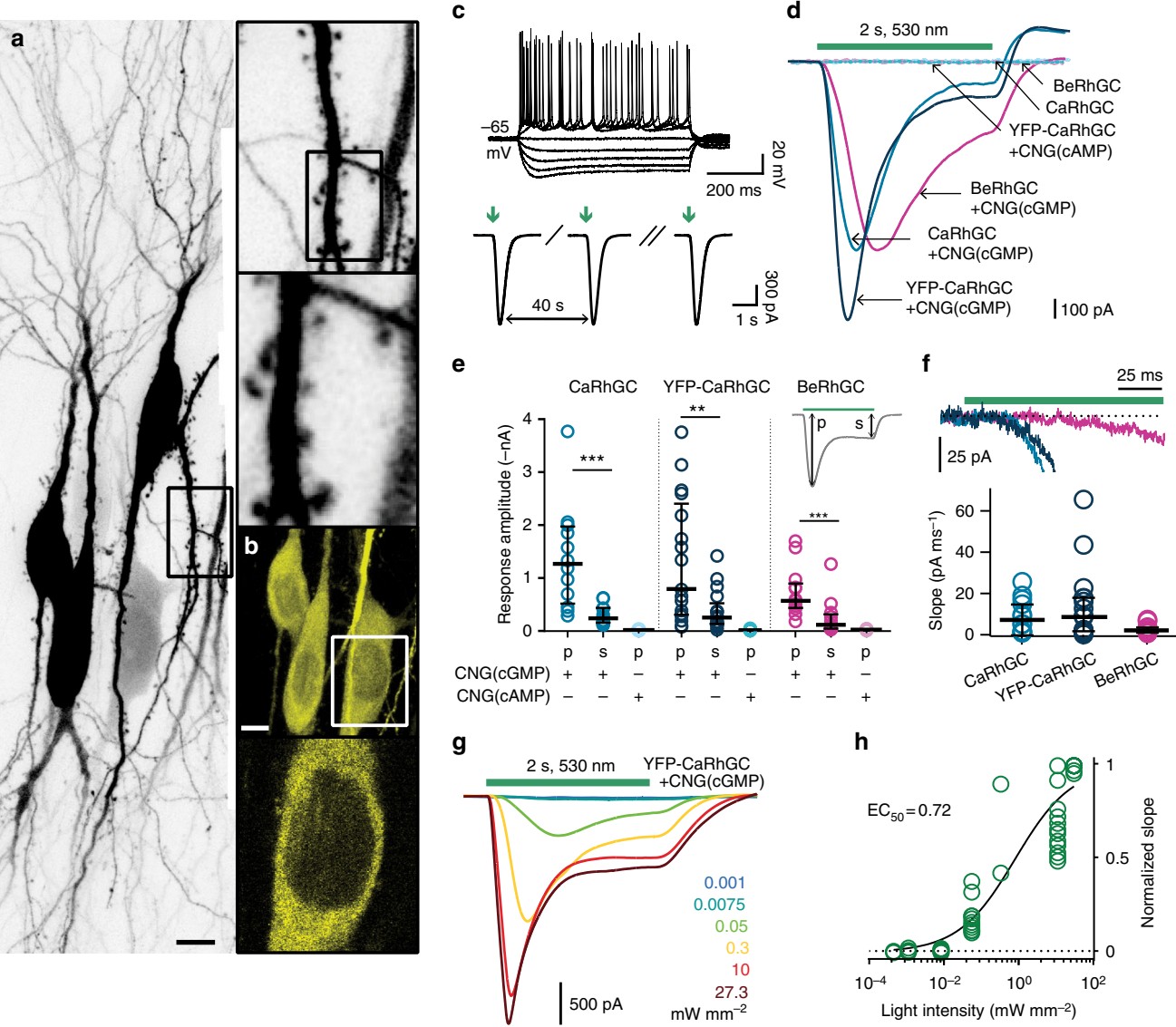

**Fig. 2** YFP-CaRhGC produces cGMP and not cAMP in hippocampal neurons. **a, b** Confocal images of neurons 8 days after electroporation with DNA encoding YFP-CaRhGC and mtSapphire; **a** maximum intensity projection, excitation 405 nm (mtSapphire); **b** top: maximum projection, excitation 515 nm (YFP); bottom: single plane. scale bars 10 μm. **c** Top: Whole-cell response to current injections from −400 pA to 400 pA in 100 pA steps. Bottom: The first, second and fifth currents evoked by repeated green light flashes (530 nm, 0.3 mW mm$^{-2}$, 100 ms, inter-stimulus interval 40 s). **d** Sample currents evoked by a 2 s green light pulse (530 nm, 27.3 mW mm$^{-2}$) in neurons expressing YFP-CaRhGC, CaRhGC or the guanylyl cyclase from *Blastocladiella emersonii* (BeRhGC) together with CNG(cGMP) or the cAMP-sensitive CNGA2 (CNG(cAMP)) channels. Green bar: light application, 2 s. **e** Peak photocurrents (p) and the sustained response (s) recorded from neurons expressing YFP-CaRhGC, CaRhGC, or BeRhGC and one of the CNG channels. Shown are individual data points, median and 25–75% interquartile range, *n*'s = 12, 12, 4, 17, 17, 6, 13, 13, 7 left to right; ***$p = 0.0001$, $p = 0.0006$, **$p = 0.009$; Mann–Whitney test, peak vs sustained response of YFP-Ca/Ca/BeRhGC + CNG(cGMP). Median peak current YFP-CaRhGC −0.79 nA, CaRhGC −1.3 nA, BeRhGC −0.6 nA; median sustained current YFP-CaRhGC −0.25 nA, CaRhGC −0.24 nA, BeRhGC −0.11 nA. **f** Detail of photocurrent onset from neurons expressing YFP-CaRhGC, CaRhGC or BeRhGC, and CNG(cGMP). Graph shows individual data points, median and interquartile range, *n*'s = 12, 17, 12. median slope YFP-CaRhGC = −8.7 pA ms$^{-1}$, CaRhGC = −7.2 pA ms$^{-1}$, BeRhGC = −2.2 pA ms$^{-1}$. **g** Sample currents recorded from a neuron expressing YFP-CaRhGC + CNG(cGMP) when stimulated with green light. **h** Light intensity-response relationship for YFP-CaRhGC + CNG(cGMP) fitted with a quadratic equation. Photocurrents were normalized to the maximum current recorded for each neuron. *n* = 17. RhGC DNA was electroporated at 10 ng μl$^{-1}$, CNG channel DNA at 25 ng μl$^{-1}$ and mtSapphire DNA at 5 ng μl$^{-1}$

shifted K$_{600}$-like intermediate that reverts via two blue-shifted states L1$_{450}$, L2$_{450}$ and the putative signaling state, M$_{380}$, back to the dark state D$_{540}$ (Fig. 3d).

**Enzymatic characterization of solubilized RhGCs and CaGC.** The high photo-stability of CaRh encouraged us to purify recombinant full-length CaRhGC from insect cells and to determine the kinetic parameters of the enzyme (Fig. 4a). A $K_M$ value

of 6.1 mM, $v_{max}$ of 821 μmol cGMP min$^{-1}$ μmol$_{protein}$$^{-1}$ and a $k_{cat}$ of 410 min$^{-1}$ was determined at pH 7.5 where the protein is most active (Fig. 4b, Table 1). The light intensity used was below the EC$_{50}$ determined from the oocyte recordings. For saturating light intensities, a higher $v_{max}$ is expected. For illuminated BeRhGC the $K_M$, maximal velocity, and turnover ($K_M = 0.92$ mM, $v_{max} = 129$ μmol cGMP min$^{-1}$ μmol$_{protein}$$^{-1}$, $k_{cat} = 64$ min$^{-1}$) were all lower than those measured for CaRhGC

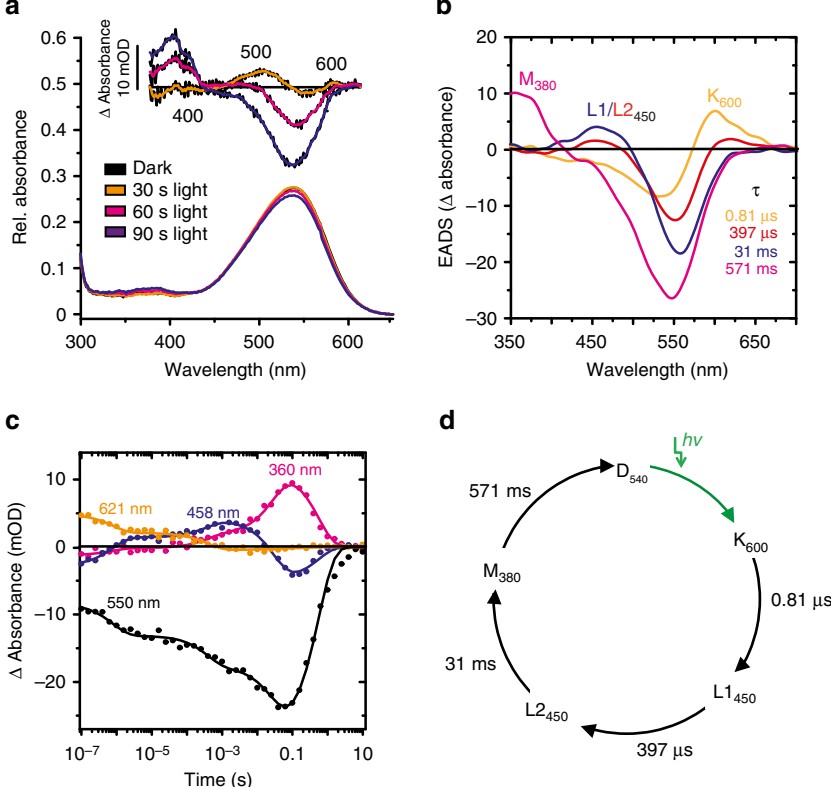

**Fig. 3** Spectroscopic analysis of purified CaRh. **a** UV–Vis spectrum of CaRh (aa 1–396) in detergent at pH 7.5 before (black) and after illumination (30–90 s, 530 nm, 0.54 mW mm$^{-2}$). Inset: light–dark difference spectra. **b** CaRh photocycle intermediates, depicted by capitals and absorption maxima, were identified by absorption changes (evolutionary-associated difference spectra (EADS)) after a short laser flash (10 ns, 532 nm, 15 mW). EADS and their life times (tau) are derived from a global fit routine. **c** Temporal dependence of photo-intermediates determined from absorbance changes at the depicted wavelengths, which were identified in **b**, plotted against time. **d** Proposed photocycle model of CaRh, details are described in the text. (Black arrows indicate thermal conversion)

(Table 1, Supplementary Fig. 4). In contrast to membrane embedded protein (Fig. 1e, Table 2), for both purified RhGCs a significant dark activity was detected and the cGMP production increased linearly with substrate concentration (Fig. 4a, Supplementary Fig. 4), suggesting a destabilizing effect of the detergent.

Next, we purified the truncated His-tagged guanylyl cyclase, CaGC, from *E. coli*. Addition of GTP/Mn$^{2+}$ revealed the cyclase domain to be constitutively active (Fig. 4c, $K_M = 5.8$ mM, $v_{max} = 136$ μmol min$^{-1}$ μmol$_{protein}^{-1}$, $k_{cat} = 68$ min$^{-1}$, maximal activity around pH 7.5, Table 1, Fig. 4d). Purified BeGC was also constitutively active (Table 1, Supplementary Fig. 5).

**Generation of rhodopsin-adenylyl cyclases.** Due to the importance of cAMP signaling and the demand for optogenetic tools that allow fast spatio-temporal control of cAMP, we sought to swap the substrate selectivity from GTP to ATP. Sequence comparison of BeRhGC and CaRhGC to other type III nucleotidyl cyclases indicated that the glutamate E497 should form a hydrogen bond with the exocyclic 2-amino group, and N1 of the guanine base and C566 should coordinate the C6-ketogroup[18] (Fig. 5a, Supplementary Fig. 6). In adenylyl cyclases, these positions are held by a lysine and an aspartate (or threonine), which anchor the adenine base through interactions with the ring nitrogen N1 and the amino group N6, respectively[19]. Similar to previous studies (e.g., refs. [18,20]), we mutated E497 to K and C566 to D and expressed the N-terminal YFP-tagged constructs in oocytes. After 3 days, oocytes were illuminated or kept in darkness, and cyclic nucleotide concentrations in lysates were quantified by ELISA using the YFP fluorescence as expression marker.

The mutations indeed changed both BeRhGC and CaRhGC into adenylyl cyclases, which we named BeRhAC and CaRhAC respectively (alternative naming: CyclOp-PACs). Similar to Trieu et al.[14], we found that in the dark BeRhAC increased resting cAMP 5× (18.5 ± 2.4 pmol/oocyte vs. 3.2 ± 1.1 pmol/oocyte non-injected oocytes) and upon illumination cAMP only increased a further 9× (Fig. 5b). In oocytes expressing CaRhAC, the dark cAMP was only 2.6× higher than in the non-injected oocytes (dark 7.9 ± 2.6 pmol/oocyte) and cAMP increased 31× when exposed to light (150 ± 28.2 pmol/oocyte, Fig. 5b). For both RhACs, no cGMP increase was detected (Fig. 5c). To reduce the dark activity of BeRhAC, we mimicked the nucleotide binding pocket of membrane anchored adenylyl cyclases (tmAC C2) by introducing four additional point mutations between aa 564 and 568 (Fig. 5a). Dark activity was not detected in oocytes expressing the E497K, 564-QYDIW-568 variants BeRhAC-6× and CaRhAC-6×, but light-induced cAMP production was reduced as well (Fig. 5b).

**Characterization of the adenylyl cyclases.** The purified CaAC domain had an activity similar to CaGC (Table 1, Supplementary Fig. 7). GTP did not serve as a substrate but was able to antagonize the production of cAMP (Supplementary Fig. 7). As for the GCs, we observed that, detergent solubilized full-length CaRhAC was de-stabilized, showing substantial dark activity and multiple bands on protein immunoblots (Supplementary Fig. 8). We therefore used membranes from oocytes expressing the YFP-tagged RhACs and CaRhGC for further in vitro enzymatic characterization. N-terminally tagged YFP-CaRhGC had lower

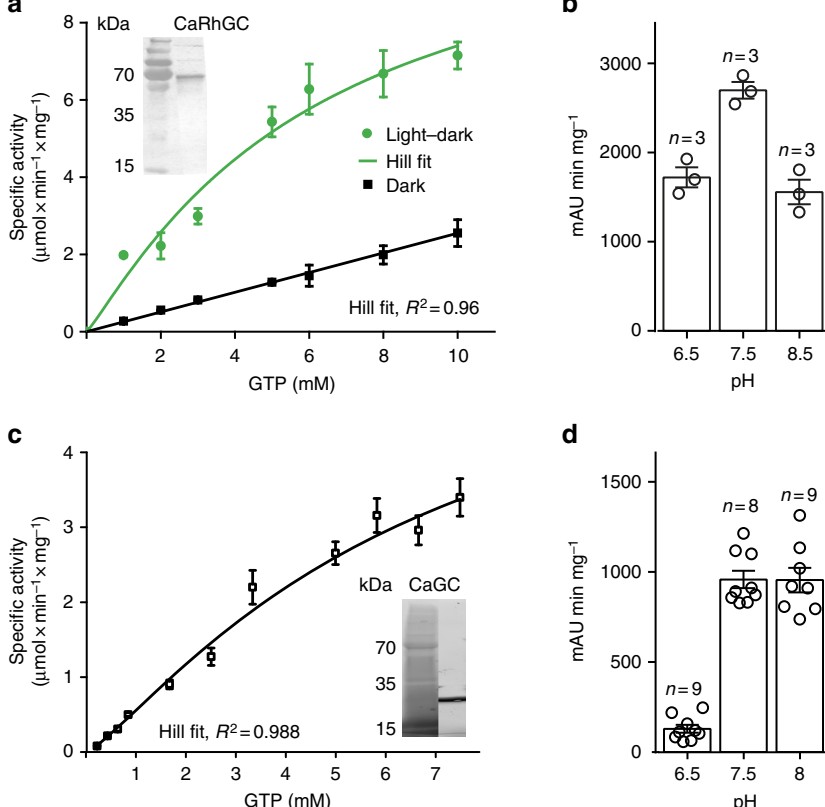

**Fig. 4** Enzymatic characterization of CaRhGC and truncated CaGC. **a** Specific activity of detergent-purified CaRhGC in light (522 nm, 0.01 mW mm$^{-2}$, green trace, dark activity subtracted) and darkness (black trace), determined at increasing GTP concentrations (0.5–10 mM GTP/Mn$^{2+}$, pH = 7.5, $n$ = 3). cGMP was quantified using RP-HPLC, a Hill fit was applied to the illuminated data (dark activity subtracted). Inset shows immunoblot (anti-his) of purified CaRhGC (MW = 71 kDa). **b** pH dependence of CaRhGC. cGMP peak areas (mAU*min) per mg protein were quantified after illumination of CaRhGC for 5 min (522 nm, 1 μW mm$^{-2}$, 1 mM GTP/Mn$^{2+}$) at the indicated pHs. **c** Specific activity of purified truncated guanylyl cyclase (CaGC) in the presence of increasing substrate concentrations (0.2–7 mM GTP/Mn$^{2+}$, pH = 7.5, $n$ = 3, data were fitted according to Hill). The inset shows the SDS gel of purified CaGC (MW = 21.5 kDa). **d** pH dependence of CaGC. cGMP peak areas (mAU*min) per mg protein were quantified after 5 min incubation of CaGC with 1 mM GTP/Mn$^{2+}$ at the indicated pHs. Bar graphs show means ± s.e.m.

### Table 1 Enzymatic parameters of purified RhGCs and truncated cyclases

| | BeRhGC (light–dark) | BeGC | CaRhGC (light–dark) | CaGC | CaAC (E497K, C566D) |
|---|---|---|---|---|---|
| Substrate | GTP | GTP | GTP | GTP | ATP |
| Hill fit $R^2$ | 0.98 | 0.99 | 0.96 | 0.98 | 0.97 |
| $K_M$ (mM) | 0.92 ± 0.27 | 2.16 ± 0.30 | 6.1 ± 5.59 | 5.78 ± 2.04 | 6.09 ± 1.74 |
| $n$ | 1 ± 0.18 | 1.72 ± 0.12 | 1.13 ± 0.42 | 1.22 ± 0.08 | 1.29 ± 0.06 |
| $v_{max}$ (cNMP (μmol min$^{-1}$ mg$_{protein}$$^{-1}$)) | 1.82 ± 0.19 | 1.89 ± 0.18 | 11.64 ± 5.67 | 6.30 ± 1.54 | 5.64 ± 1.24 |
| $v_{max}$ (cNMP (μmol min$^{-1}$ μmol$_{protein}$$^{-1}$)) | 128.5 ± 13.4 | 40.67 ± 3.87 | 821.8 ± 400.3 | 135.59 ± 33.14 | 121.38 ± 26.68 |
| $k_{cat}$ (min$^{-1}$) | 64.3 | 20.34 | 410.9 | 67.80 | 60.69 |
| $k_{cat}$ (s$^{-1}$) | 1.1 | 0.34 | 6.85 | 1.13 | 1.01 |
| $k_{cat}/K_M$ (s$^{-1}$ mM$^{-1}$) | 1.2 | 0.16 | 1.12 | 0.20 | 0.17 |
| Molecular weight (kDa) | 70.6 | 21.5 | 70.6 | 21.5 | 21.5 |
| Illumination | 522 nm, 0.010 mW mm$^{-2}$ | | 522 nm, 0.010 mW mm$^{-2}$ | | |

Values are mean ± sem

cGMP turnover in the dark than CaRhGC-YFP as expected (Table 2). In the light, cGMP turnover increased >1000×, which is comparable to the activity measured for BeRhGC (BeCyclOp) using the same assay[12]. For YFP-RhAC from both organisms, a light-driven cAMP turnover of ~40 min$^{-1}$ and a light/dark activity ratio of ~200 was determined (Table 2). The lower activity of the RhACs compared to RhGCs and the lower dark activity of YFP-CaRhAC compared to YFP-BeRhAC confirmed the results from oocyte lysates (Table 2, Fig. 5b). The oocyte membrane

assay also confirmed the reduced dark activities of YFP-RhACs-6× and increased light/dark activity ratios. Cyclic GMP was not produced by any RhAC mutants and conversely, cAMP was not produced by CaRhGCs (Table 2).

We tested several versions of CaRhAC in hippocampal neurons. YFP-CaRhAC with only the E497K/C566D mutations and no C-terminal mycHis tag was superior, producing light-induced cAMP mediated currents in all transfected neurons (Fig. 6, Supplementary Table 3, Supplementary Figs. 3, 9, 10).

**Table 2 In vitro production of cAMP and cGMP by RhACs in oocyte membranes**

| | cGMP turnover in darkness (min$^{-1}$) | cGMP turnover in light (min$^{-1}$) | L/D | cAMP turnover in Dark (min$^{-1}$) | cAMP turnover in Light (min$^{-1}$) | L/D |
|---|---|---|---|---|---|---|
| YFP-CaRhGC (20 °C) | 0.028 ± 0.007 | 71 ± 10.7 | 2500 | ND | ND | — |
| CaRhGC-YFP (20 °C) | 0.11 ± 0.04 | 120 ± 9.3 | 1100 | ND | ND | — |
| YFP-BeRhAC (20 °C) | ND | ND | — | 0.19 ± 0.01 | 42.1 ± 0.6 | 220 |
| YFP-CaRhAC (20 °C) | ND | ND | — | 0.14 ± 0.01 | 39.4 ± 5.6 | 280 |
| YFP-BeRhAC-6 × (20 °C) | ND | ND | — | <0.001 | 0.7 ± 0.13 | >700 |
| YFP-BeRhAC-6 × (37 °C) | ND | ND | — | <0.001 | 3.2 ± 0.8 | >3200 |
| YFP-CaRhAC-6 × (20 °C) | ND | ND | — | 0.001 ± 0.0001 | 0.74 ± 0.08 | 740 |
| YFP-CaRhAC-6 × (37 °C) | ND | ND | — | 0.016 ± 0.005 | 1.72 ± 0.36 | 108 |

Values are mean ± standard deviation
RhAC E497K C566D, RhAC-6× E497K 564QYDIW568, ND not detectable

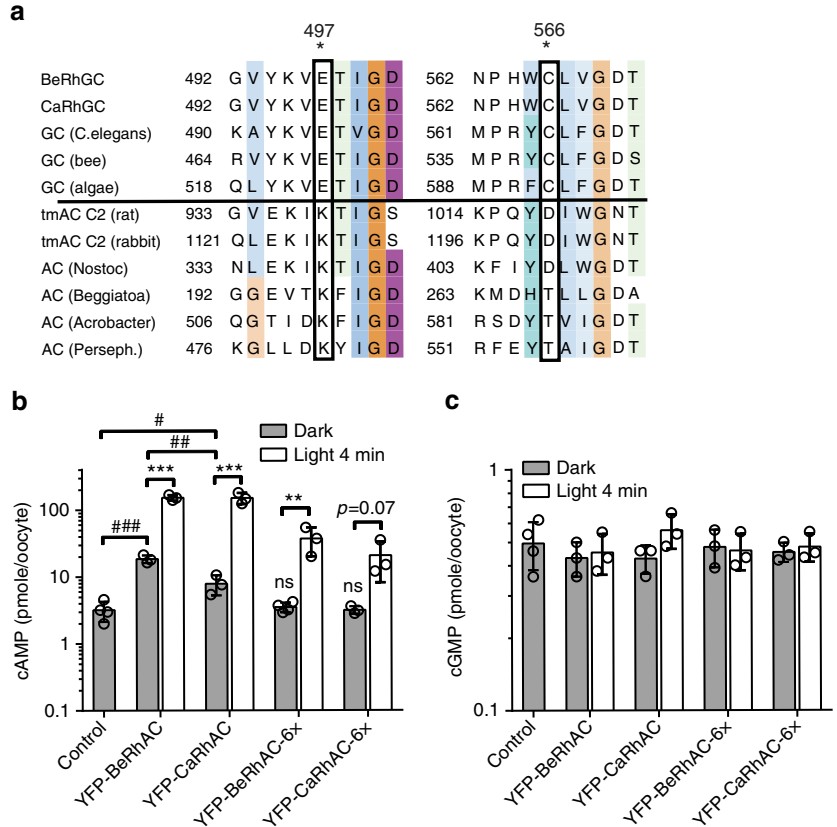

**Fig. 5** Generation and characterization of adenylyl cyclases. **a** Sequence alignment of adenylyl cyclases (ACs) and guanylyl cyclases (GCs) from various organisms (full alignment and accession numbers in Supplementary Figure 6) shows key residues (*), involved in nucleotide binding, which differ between ACs and GCs. Insertion of the double mutation E497K, C566D-generated Ca/BeRhACs. Four additional mutations (564-QYDIW-568) were inserted in Ca/BeRhACs-6×. Enzymatic specificities of the RhACs were determined via ELISA-based quantification of cAMP (**b**) and cGMP (**c**) within oocytes, expressing the N-terminal YFP-tagged constructs as indicated. Oocytes were kept in the dark or illuminated (light 4 min, 532 nm, 0.3 mW mm$^{-2}$) immediately before lysis. Bar graphs show data as means ± s.d., $n = 3$ samples of 5 oocytes each, ***$p = 8 \times 10^{-5}$, $9 \times 10^{-4}$, **$p = 0.01$, unpaired $t$-test light vs dark; ###$p < 0.0001$, ##$p = 0.001$, #$p = 0.02$, ns = not significant from control one-way ANOVA (dark conditions $p < 0.0001$) followed by Tukey's multiple comparisons, for clarity not all comparisions are shown. Control = non-injected oocytes

Neurons expressing YFP-BeRhAC-6× or YFP-CaRhAC-6× frequently had multiple nuclei and were difficult to record from. Hippocampal neurons expressing YFP-CaRhAC had normal morphology and membrane properties (Fig. 6a–c). The YFP fluorescence appeared to be associated with the plasma membrane and intracellular membranes (Fig. 6b). Green flashes repeatedly evoked transient currents through the cAMP-sensitive channels. The rise and decay of the currents was slower than for the YFP-CaRhGC evoked currents in Fig. 2c (Fig. 6d, e (inset),

Supplementary Table 3, Supplementary Figs. 3d, 9b, d). Neurons expressing YFP-CaRhAC alone or together with CNG(cGMP) had no or small photocurrents (Fig. 6e, f, Supplementary Table 3, Supplementary Fig. 9). As hippocampal neurons have endogenous hyperpolarization-activated and cyclic nucleotide-gated channels (HCN)[21], at least part of these residual currents are likely due to activation of endogenous channels by cAMP. The cAMP induced photocurrents were light intensity-dependent with an EC$_{50}$ of 0.6 mW mm$^{-2}$ (Fig. 6g, h). As expected, all the

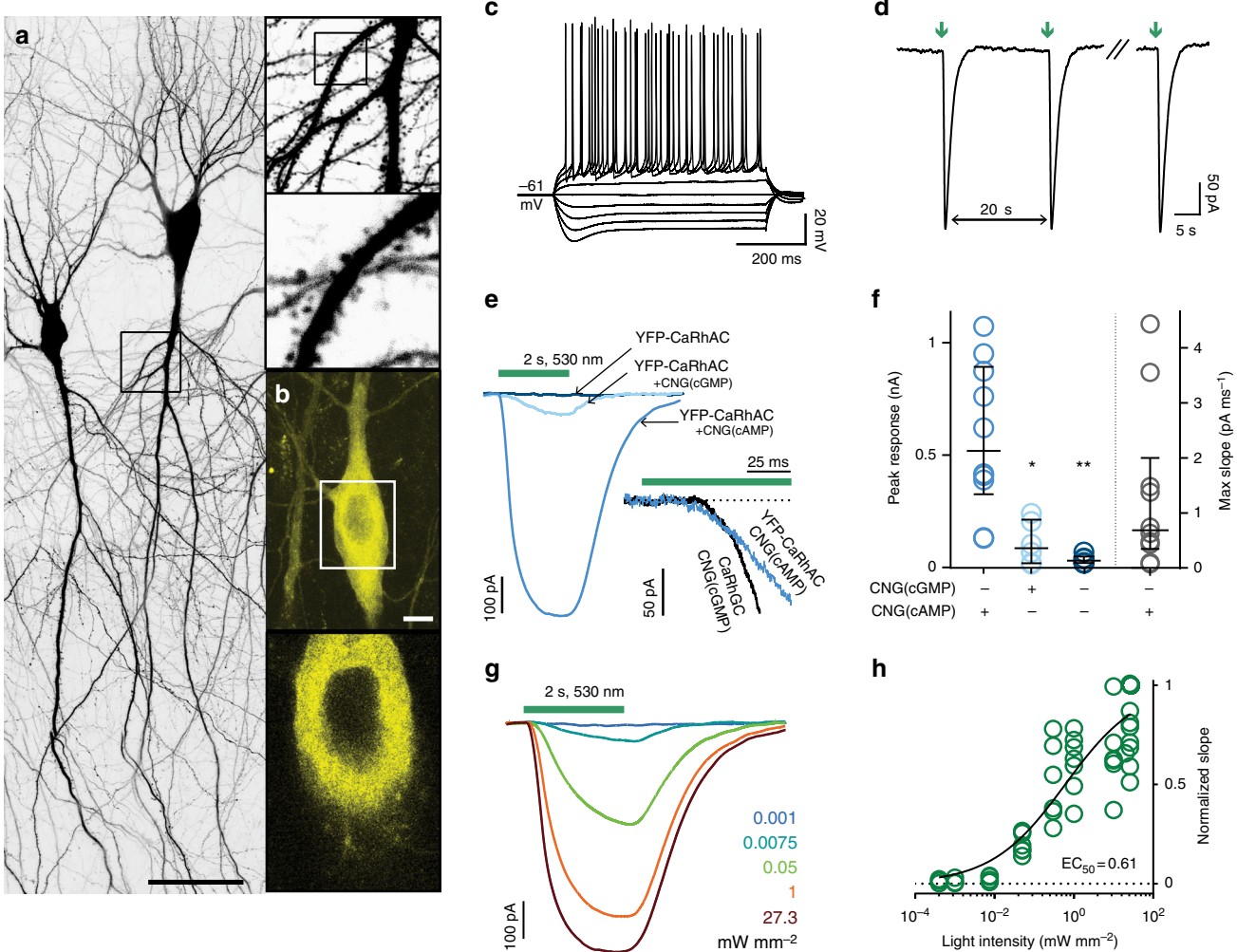

**Fig. 6** YFP-CaRhAC rapidly produces cAMP in hippocampal neurons. Two photon images of a neuron expressing YFP-CaRhAC (YFP-BeRhGC E497K C566D) and mtSapphire (**a**) or YFP-CaRhAC alone (**b**); **a** maximum intensity projection, excitation 800 nm, scale bar 50 μm; **b** top: maximum projection, excitation 950 nm; bottom: single plane YFP fluorescence 950 nm of same neuron. scale bar 10 μm. **c** Whole-cell response to current injections of a hippocampal neuron expressing YFP-CaRhAC plus the cAMP-sensitive CNGA2 channel (CNG(cAMP) C460W/E583M). Current injections from -400 pA to 400 pA in 100 pA steps. **d** The first, second and fifth currents evoked by repeated green light flashes (530 nm, 0.3 mW mm$^{-2}$, 100 ms, interval 20 s). **e** Sample currents evoked by strong green light (27.3 mW mm$^{-2}$) in neurons expressing YFP-CaRhAC together with CNG(cGMP), CNG(cAMP) or alone. Green bar: light application, 2 s. Insert shows the onset and initial slope of the YFP-CaRhAC + CNG(cAMP) photocurrents in comparison to photocurrents from a CaRhGC + CNG(cGMP) expressing neuron. **f** Comparison of the maximum peak or slope of photocurrents recorded from neurons expressing YFP-CaRhAC and one of the CNG channels or by itself. Shown are individual data points, median and interquartile range, n's = 10, 6, 6, 10 left to right. *p = 0.023, **p = 0.0016, Kruskal–Wallis test vs YFP-CaRhAC + CNG(cAMP). **g** Sample currents recorded from a neuron expressing YFP-CaRhAC + CNG (cAMP) when stimulated with green light of different intensity. **h** Light intensity-response relationship for YFP-CaRhAC + CNG(cAMP) fitted with a quadratic equation. Photocurrents were normalized to the maximum current recorded for each neuron. n = 8. DNA encoding rhodopsin-adenylyl cyclases was electroporated at 25 ng μl$^{-1}$, CNG channel DNA at 25 ng μl$^{-1}$ and mtSapphire DNA at 5 ng μl$^{-1}$

constructs retained the highest sensitivity to green light in neurons (Supplementary Fig. 11).

**Crystal structure of the adenylyl cyclase domain.** To gain structural information about the class of enzyme rhodopsin and the nucleotide binding mode we tried to crystallize both the wild-type full-length and the isolated enzyme domains of CaRhGC and CaRhAC in presence of NTP analogs. While crystallization of the full-length CaRhGC failed, we produced highly diffracting crystals of the GC domain (aa 443–626, molecular replacement based on PDB: 4P2F). Similar to Kumar et al.[22], our GC structures either revealed a monomeric or a non-functional dimeric arrangement, connected by an artificial disulfide bridge. Since the

structures did not contain a bound substrate analog, they were not considered further. Crystals of CaAC in complex with ATPαS diffracted to a maximum resolution of 2.25 Å. In contrast to our GC structures, CaAC crystallized as a homodimer in an anti-parallel orientation forming two symmetric catalytic sites within the protein–protein interface, as expected from other adenylyl[19,23] and guanylyl cyclase[24–26] structures. The structure was solved by molecular replacement using our high-resolution structure (1.2 Å) of monomeric CaGC, which allowed us to refine the structure to an R factor of 18.2 % and free R factor of 22.4 % (Table 3). The CaAC exhibits the classical nucleotidyl cyclase type III fold with a central 7 stranded β-sheet shielded by 3 helices. The nomenclature of the secondary structure elements are used according to Zhang et al.[27] (Supplementary Fig. 12b).

**Table 3 Data collection and refinement statistics**

|  | CaAC (PDB-ID: 5OYH) |
|---|---|
| *Data collection* |  |
| Space group | I41 |
| Cell dimensions |  |
| *a, b, c* (Å) | 193.28, 193.28, 225.50 |
| *α, β, γ* (°) | 90, 90, 90 |
| Resolution (Å) | 45.9–2.249 (2.33–2.249)[a] |
| Unique reflections | 194,816 (19,386) |
| $R_{merge}$ | 0.1151 (1.056) |
| $CC_{1/2}$ | 0.999 (0.703) |
| $I/\sigma I$ | 15.35 (1.92) |
| Completeness (%) | 100 (100) |
| Redundancy | 7.8 (7.9) |
| Wilson *B*-factor | 36.01 |
| *Refinement* |  |
| Resolution (Å) | 2.25 |
| No. of reflections | 194,681 (19,371) |
| $R_{work}/R_{free}$(%) | 18.2/22.4 |
| No. of atoms | 24,979 |
| Protein | 226,62 |
| Ligand/ion | 602 |
| Water | 1715 |
| *B*-factors | 40.34 |
| Protein | 39.61 |
| Ligand/ion | 56.47 |
| Water | 44.28 |
| R.m.s. deviations |  |
| Bond lengths (Å) | 0.009 |
| Bond angles (°) | 0.9 |
| Ramachandran Plot |  |
| Ramachandran favored (%) | 98 |
| Ramachandran allowed (%) | 1.7 |
| Ramachandran outliers (%) | 0 |

[a]Values in parentheses are for highest-resolution shell

Within the crystal lattice the CaAC homodimers formed an intertwined helical superstructure, an unusual packing that leads to a large unit cell (8426 nm³) with 8 homodimers per asymmetric unit (Fig. 7a). The dimer composed by chain A/B (Fig. 7b) shows the smallest mean overall B-factor for the binding pocket and therefore serves as main basis for the structural description given below.

The monomers of one particular homodimer superimpose very well (<0.2 Å RMSD between the Cα positions) but there is a prominent asymmetry found for the location of the two β4/5 loops (residue L558 to P563, Fig. 7b, c), a region supposed to be involved in intramolecular signal transduction of the flavin-based photo-activated cyclase bPAC (see Discussion for details). In six of eight modeled dimers of CaAC, steric restriction by the crystal packing lead to a conformation where one loop is pointing away from the central core (loop-distal), while the other is oriented towards the core (loop-proximal) (Fig. 7c). The electron density found for the proximal loop position is less well-defined and only the protein backbone could be modeled reliably. In the remaining two dimers (chains M/N and O/P), in which these loops are not stabilized, electron density is absent indicating the high conformational flexibility of this region.

**The active site of CaAC.** In all CaAC homodimers a well-defined electron density is present in both substrate-binding sites, which allowed modeling the ATP analog ATPαS with full occupancy (Fig. 7d). Both diastereomers of the substrate analog with sulfur substituting the Pα oxygen either in Rp or Sp position (Supplementary Fig. 12a) were present during crystallogenesis and

biochemical analysis showed a similar inhibitory potential for both (Supplementary Fig. 13). The refinement of ATP-Sp-αS within the binding pocket lowers the B-factors obtained for the sulfur position to values more comparable to the overall molecule in all 16 ligands and was therefore included in the model. As in other AC structures, residues that anchor the adenine and phosphate tail of ATPαS in the binding pocket belong to different monomers of the dimeric CaAC. In the following, residues belonging to the monomer that bind the phosphate tail are marked (*).

Similar to other type III cyclases, 7 conserved residues in CaAC provide ligand binding and are mainly involved in formation of cAMP and PPi from ATP (indicated in Supplementary Fig. 6)[19]. ATP cleavage and cyclization is considered to follow an intramolecular nucleophilic substitution ($S_N2$), which is initiated through the attack of ribose-3′-OH oxygen at Pα, resulting in a negatively charged α-phosphate during the transition state[19,23,28].

The adenine base is held within the hydrophobic cleft of the nucleotide binding pocket defined by F455, L504, L567, V568, V572, V496, I499 and I499*, while the mutated residues E497K and C566D anchor the base via hydrogen bonds (Fig. 7d, e). In particular, D566 together with the backbone oxygen of L567 function as hydrogen bond acceptors for the amino group of the adenine base while K497 serves as donor for the ring nitrogen N1. An additional water-mediated interaction ties purine N7 to the conserved N573 (α4) and G569 (α4). This observed donor/acceptor pattern would be unfavorable for a similarly positioned guanine. Although these key positions for base selectivity were identified 20 years ago, the CaAC structure explains the implications of E497K and C566D for the substrate specificity switch on a structural level. Different to most other adenylyl cyclase structures, the ribosyl moiety (2′-endo conformation) is tilted perpendicular to the purine plane and orients towards the β2/3* loop with the 2′-OH in hydrogen bond distance to D501* and the backbone carbonyl of I499*(Fig. 7d, f). Two waters are seen that are expected to stabilize this binding to β2/3* by mediating further interactions to 2′-OH and 3′-OH. Different to other adenylyl cyclases[29] the ribose ring oxygen of CaAC forms a weak hydrogen bond with S576 (α4, 3.5 Å) (Fig. 7e) and does not interact with the conserved and catalytically important N573 (α4).

In proximity to the ribose-3′-OH, two metal ions (A and B) have been detected for other adenylyl cyclases[30,31]. While ion A has been associated with the abstraction of a proton from 3′-OH-ribose and thus involved in catalysis, ion B was found to bind the substrate through coordination of Pβ-Pγ. In CaAC the metal B site is occupied by a metal ion (Fig. 7d, f), which was assigned to calcium based on the coordination number and distances, while ion A is absent. $Ca^{2+}$ is octahedrally coordinated by the conserved residues D457* and D501*, the backbone carbonyl of I458*, the oxygens of the β and γ phosphates and a less well-defined weakly bound water. Both aspartates provide only one of their oxygens to bind the metal. The second oxygen of D457* forms a further contact to R545*, whereas D501* forms a hydrogen bond with the ribose.

Apart from binding to calcium, the terminal β-γ phosphates are located close to helix α1* and forms hydrogen bonds with T462*, F461* and N460* of the adjacent loop. The γ phosphate orients towards R545*, a residues suggested to aid the exit of the formed pyrophosphate[32].

In particular, the orientation of the conserved arginine (R577 in CaAC), located on α4 (Fig. 7d, f), towards Pα was claimed to be essential to stabilize the additional negative charge of the α-phosphate after the ribose-3′-OH attack[32]. In CaAC this arginine (R577) (Fig. 7f) is found to be flexible and electron density for its side chain is not present in all protein

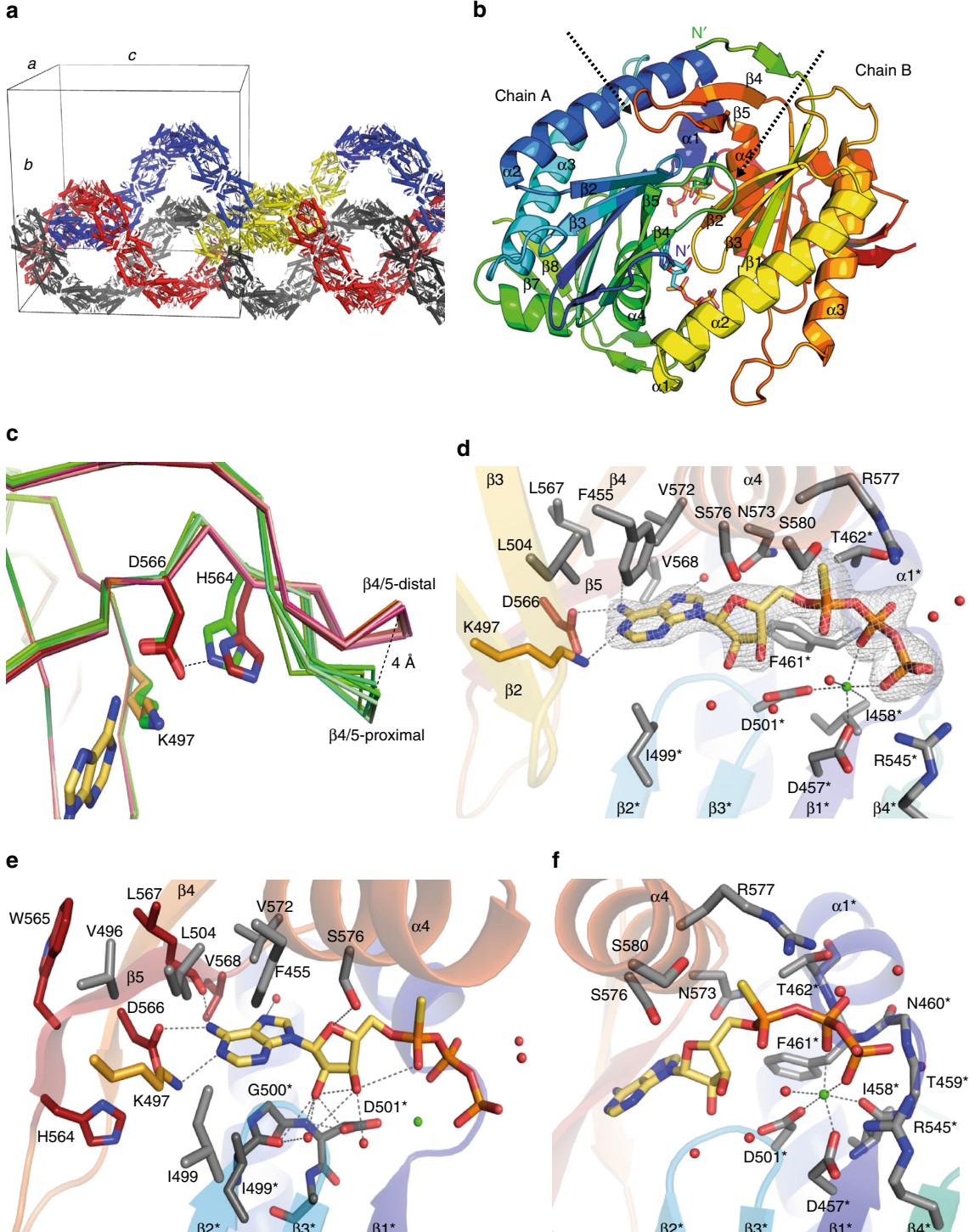

**Fig. 7** Crystal structure of CaAC in complex with ATPαS. **a** CaAC (E497K/C566D) homodimers form an intertwined helical superstructure within the crystal lattice (blue, red, gray indicate individual superhelices, yellow indicates 8 homodimers within a single asymmetric unit). **b** Each homodimer consists of two antiparallel arranged monomers and harbors 2 active sites at the dimer interface, occupied by the ATP analog: ATP-Sp-αS. Arrows indicate β4/5 loops, N-termini are labeled, chain A: blue/green, chain B: green/red. **c** Comparison of the β4/5 loop orientation in CaAC. Overlay of the six monomers of CaAC found in the loop-proximal (green) position close to the catalytic core, or the loop-distal (orange) position. The protein backbone is visualized as a ribbon with selected residues and the base head group shown as sticks. **d** CaAC active site; the adenine base and phosphate tail of ATP-Sp-αS are anchored by different monomers. Residues of the phosphate binding monomer are marked (*). The previously mutated residues E497K (orange), C566D (red) form hydrogen bonds to the adenine base. The ribose is tilted perpendicular to the adenine plane and points towards β2*, β3*. The conserved arginine, R577 (α4) orients towards Pα. The ion B site is occupied by $Ca^{2+}$, which is octahedrally coordinated among others by two conserved aspartates, D457* and D501*. The phosphate tail of ATP-Sp-αS is anchored to the α1/ β1-loop via polar interactions. mFo-DFc omit map for ATP-Sp-αS contoured at 5σ, contour level is shown as gray mesh. **e** Magnification of the adenine base binding residues. Neighboring residues of D566, situated on β5 and mutated in RhACs-6 × are colored in red. **f** Magnification of the residues, involved in nucleotide phosphate binding, coloring as in **d**. Gray dashed lines indicate hydrogen bonds or metal coordination between 2.3 Å and 3.6 Å in length

subunits. In most cases the guanidine group points towards the Pα,β anhydride bond, a position suitable for stabilizing the transition state.

## Discussion

In this study, we characterized the rhodopsin-guanylyl cyclase CaRhGC from the fungus *Catenaria anguillulae*, which belongs to a recently discovered class of enzyme rhodopsins (cyclase opsins). CaRhGC produces cGMP upon illumination with green light in oocytes and rat hippocampal neurons. As the guanylyl cyclase domain (aa 443–626) itself is active without the rhodopsin, we conclude that in full-length CaRhGC (and BeRhGC) the rhodopsin domain acts as a clamp, which keeps the cyclase inactive in darkness, and illumination releases the clamp allowing the cyclase to produce cGMP. Interestingly, the light-activated rhodopsin additionally influences the cyclase dynamics, as the RhGCs are 3–6× more active than the isolated cyclase domains (Fig. 4, Supplementary Figs. 4–5, Table 1). Possibly the activated rhodopsin stabilizes the dimeric arrangement of the enzyme or light causes other conformational changes that promote catalysis. A full understanding of the light-induced intramolecular signal transduction will require solving the structures of full-length dark-adapted and light-activated rhodopsin cyclases.

Spectroscopic analysis of the CaRh domain indicated a higher photo-stability compared to that of BeRh, which is beneficial for optogenetic approaches or biophysical studies where extended illumination protocols are required. Different to most microbial rhodopsins the prolonged M state is formed rather late in CaRh. But, in histidine kinase rhodopsin 1 (HKR1) of *Chlamydomonas* the M state is formed also rather late within 27 ms after excitation[33]. We expect that the M state (probably a late M) is the cGMP producing signaling state of CaRhGC, since pronounced structural changes occur during the M1 to M2 transition in other microbial rhodopsins as bacteriorhodopsin[34]. In addition, the median onset of photocurrents was around 25 ms in neurons expressing any of the *Catenaria* cylcases, setting an upper limit for formation of the active state. Thus, the M state or an earlier state (i.e., L2) must be the active state.

Compared to BeRhGC, the prolonged M state and associated active state of CaRhGC increases the number of molecules in the photocycle at non-saturating light intensities, which accelerates the accumulation of cGMP. This would explain (a) the 5× faster onset of the CaRhGC photocurrents and (b) the increased enzymatic activity of CaRhGC compared to BeRhGC (Table 1). The $v_{max}$ of isolated CaGC was also higher than BeGC, contributing to the improved performance of full-length CaRhGC.

Introducing the point mutations E497K/C566D into the two RhGCs successfully converted the rhodopsin-guanylyl cyclases into rhodopsin-adenylyl cyclases. Unfortunately, BeRhAC had dark activity, as recently reported[14] and the construct was poorly tolerated in neurons. CaRhAC was less active in darkness and better tolerated by neurons but was unreliable, increasing cAMP in only a third of transfected neurons when illuminated. The N-terminally tagged YFP-CaRhAC version with improved performance might be better expressed and/or membrane-targeted. Additionally, the N-terminal YFP may stabilize the dark state of the full-length protein. The reason for the increased dark activity of RhGC with C-terminal YFP remains unclear but is not caused by the proteolytic separation of the constitutively active GC domain as shown by immunodetection (Supplementary Fig. 14). Even attaching smaller mycHis tags to the C terminus had a negative impact (Supplementary Table 3).

Several other optogenetic tools exist for raising cAMP. In comparison to the soluble bPAC or euPAC, YFP-CaRhAC allows faster control of intracellular cAMP (ms vs seconds time-scale) and being a transmembrane protein, more closely mimics the endogenous transmembrane adenylyl cyclases[35]. Additionally, it may prove easier to target CaRhAC (and wt RhGCs) to specific cellular compartments, which will enable studying subcellular effects of cAMP (and cGMP)[36]. While bPAC and CaRhAC are single proteins, the G protein-coupled rhodopsins OptoXRs and JellyOps rely on activation of endogenous G proteins and endogenous transmembrane ACs to raise cAMP[37,38]. We expect that single component cyclases will less likely activate additional cNMP independent signaling pathways and that they will function in a wider variety of cell types.

In nucleotidyl cyclases, a head-to-tail arrangement of two cyclase subunits is required for catalysis. Surprisingly, the isolated GCs crystalized as monomers or unusual head-to-head dimers[22]. In contrast, CaAC crystallized as an antiparallel homodimer, which indicates a classical type III cyclase reaction mechanism. According to current knowledge (mostly derived from ACs), the reaction is initiated by substrate-binding to an open conformation of the protein mainly through interactions between the triphosphate and metal ion B[19]. The transient binding of the second, catalytic metal ion A facilitates the deprotonation of the ribose-3′-OH needed for its nucleophilic attack at Pα. It is assumed, that the base discriminating interactions and substrate specificity is achieved during catalysis and not upon substrate-binding. During cyclization the nucleotide must adopt an inline orientation of the attacking ribose-3′-O and the 'to be broken' Pα-O bond. This leads to a pentavalent transition state, in which the α-phosphate harbors an additional negative charge stabilized by ion A and the conserved arginine R577 in CaAC. Proper placement of this arginine is thought to be facilitated by protein closure based on the approach of the β7/8 loop and helix *a*1 towards the dimer center. We believe that CaAC adopts a closed conformation for two reasons: the CaAC helix α1 position resembles α1 1 in other closed-state AC structures as for example tmAC (1CJK)[31] (Supplementary Fig. 12c) and the hydrogen bond pattern of β7/8 to the neighboring helix α3 and α1* match that of closed transmembrane AC. It remains to be answered, if this closure results (a) from a ligand-induced fit and shows the enzyme before catalysis[29] or, (b) represents the product/enzyme complex[30].

The CaAC binding site is highly similar to other ligand-bound ACs with either both metal sites occupied or a single ion B. Ion A, believed to bind transiently during catalysis, is only present in closed-state structures of tmAC and cyanobacterial AC, in which the non-productive nucleotide conformation stabilizes the metal interaction. Different to most other AC structures, the ribose interacts with the β2*/3* loop (via D501*). Interestingly, this placement brings the 3′-OH close to the position of the ion A-site deduced from the structure of tmAC (Supplementary Fig. 12d). Indeed, density functional calculations (DFT)[39] of the active site based on tmAC suggest that the sugar approaches Ion A to allow deprotonation of the attacking 3′-OH. Thus, the CaAC ribose position may be of catalytic relevance.

Electron density for a single conformer, assigned as ATP-Sp-αS was found in the CaAC crystal. Since ATP-Sp-αS (but not ATP-Rp-αS) is catalyzed by other ACs[30], finding this epimer in an AC crystal structure is unexpected. In other ACs, stereospecicic inhibition by ATP-Rp-αS is attributed to a deviation from the inline arrangement of the ribose-3′-O and the 'to be broken' Pα-O bond, which prevents catalysis. In CaAC the epimer ATP-Sp-αS is an equally potent inhibitor (Supplementary Fig. 13) and the ribose-3′-OH and Pα−O are not arranged inline, similar to other inhibitor bound AC structures. The contribution of the sulfur to the nucleotide conformation remains uncertain.

An open question is how the rhodopsin photon absorption results in cyclase activity. Similar to other rhodopsins, we assume

a light-induced conformational change, which is transmitted via the coiled-coil domain to the cyclase. Recently the structures of two bacterial light-activated type III adenylyl cyclases (OaPAC[40,41] and bPAC[42]) were solved and the studies addressed how light-induced conformational changes within the BLUF domain (sensors of blue light using flavin adenine dinucleotide) are propagated via the coiled-coil domain to regulate the AC domain. Based on dark-adapted and light-activated OaPAC structures[40,41], Ohki et al. suggested that intramolecular signal transduction is based on protein vibrations and minimal differences in atomic positions. In contrast, Lindner et al.[42] proposed a light-induced β4/5 loop movement within the AC domain of bPAC, opening the catalytic cleft and aligning the substrate-binding residues, which are necessary for catalysis. Although superimposition of bPAC and CaAC results in a steric clash of the coiled-coil helices (bPAC) with β4/5 of CaAC (Supplementary Fig. 12e), we assume by analogy a central role for β4/5 in signal transmission. According to the crystallographic model, this flexible loop directly precedes β5, which forms the main protein contacts to the nucleotide base. Thus, movement of the β4/5 loop is expected to directly affect base binding and consequent substrate turnover. Indeed, the two orientations of this loop present in the CaAC structure show slightly different hydrogen bonding to D566 due to a shift of H564 (Fig. 7c), nevertheless base coordination is unchanged. Additional mutations near position 566 in RhACs-6× decrease both dark- and light-induced enzymatic activity, indicating the importance of this region for enzyme activity. Although our study provides structural insights into the recently discovered class of enzyme rhodopsins, the structure of the full-length protein will be crucial for a detailed understanding of the intramolecular photoactivation mechanism.

## Methods

**Animal experiments**. All animal experiments were approved by the local authorities.

Berlin: All experiments were conducted in accordance with the guideline given by Landesamt für Gesundheit und Soziales Berlin and were approved by this authority.

Würzburg: *Xenopus laevis* surgery for oocytes in Wuerzburg was under License #70/14 from Landratsamt Wuerzburg Veterinaeramt.

Hamburg: Rats were housed and bred at the University Medical Center Hamburg animal facility. All procedures were performed according to protocols approved by the Animal Research Ethics Board (AREB) and the Institutional Animal Care and Use committee of the City of Hamburg.

**Molecular biology**. Both rhodopsin-guanylyl cyclase sequences (1–626 residues) from *Catenaria anguillulae* (Ca) (gb: MF939579) and *Blastocladiella emersonii* (Be) (gb: AIC07007.1) were ordered from GenScript and the RhAC variants (wtih 2–6 amino acids mutated) were generated with the help of a quickchange PCR reaction. An overview of the used constructs is found in Supplementary Table 4.

For electrophysiological measurements in *Xenopus* oocytes, RhGCs and RhACs (Be/Ca) were subcloned via *Bam*HI and *Hin*dIII into pGEM (Promega), (#2, Supplementary Table 4).

For Elisa assays, the N-terminally YFP-tagged versions were cloned by *Xho*I and *Hin*dIII after the YFP coding sequence in the pGEM vector. The C-terminal YFP-tagged versions were cloned by BamHI and XhoI before the YFP coding sequence in the pGEM vector (#3–7, 12, Supplementary Table 4).

For expression and purification of the *Catenaria* rhodopsin domain (1–396) and the CaRhGC/CaRhAC full-length constructs, DNA sequences were subcloned into the pFastBac plasmid (Thermo Fisher Scientific) at the BamHI and HindIII restriction sites. All constructs contained a C-terminal TEV-cleavage site followed by a polyhistidine tag (8HIS) (#18, #22-23, Supplementary Table 4).

For expression and purification of the soluble wild-type/mutated cyclase domains (residue 443–626), cyclases were cloned via *Nde*I and *Xho*I into pET21a+ (Novagen), 5′ upstream of a 6×His-tag (#19-21, Supplementary Table 4).

For electrophysiology in hippocampal neurons, BeRhAC was generated from pAAV-Syn-hbeRhGC-2A-tDimer (Addgene plasmid #66779) by QuickChange PCR (primers: 5′-GTGTATAAAGTCAAGACAATCGGCG-3′, 5′-CGCCGATTG TCTTGACTTTATACAC-3′; 5′-AACCCACACTGGGATCTGGTGGGAGA-3′, 5′-TCTCCCACCAGATCCCAGTGTGGGTT-3′), (#8, Supplementary Table 4).

C-terminal tagged (myc and His) CaRhAC/CaRhGC constructs were obtained by cloning from pGEM constructs (used in oocyte assays) into pAAV-hSyn1-mRuby2-GSG-P2A-GCaMP6s-WPRE-pA (Addgene plasmid # 50942) via *Sac*I/ *Hin*dIII and *Pas*I/*Hin*dIII, respectively (primers: 5′-CTGAGCTCATGTCTATGAA AGATAAAGA-3′, 5′-TAGTGGTAACCAGATC-3′, (#14–15, Supplementary Table 4).

The untagged variants (CaRhGC and CaRhAC in text) were generated from the pAAV-Syn-mRuby2-GSG-2A-CaRhGC/CaRhAC variants into pAAV-Syn-hbeRhGC-2A-tDimer by PCR and restriction with *Eco*RI and Acc65I (primers: 5′-TCGAATTCGGCAATAATGTCTATGTCTATGAAAGAT-3′, 5′-CGGGTACCGA CTTTCTAGCGGTCACCCAATATG-3′, (#10-11, Supplementary Table 4).

YFP-tagged CaRhAC/CaRhGC (mouse codons) were cloned from pGEM-YFP-CaRhAC used in oocyte recordings into pAAV-Syn-YFP-CaRhAC via *Eco*RI and Acc65I (primers: 5′-TAGTGGTAACCAGATC-3′, 5′-CACTGGAGCTATCAACGGAG-3′), (#12-13, Supplementary Table 4).

Restriction enzymes and Pfu DNA Polymerase were acquired from Thermo Fisher Biosciences, primers were custom made at Eurofins. Macherey-Nagel MN Center's NucleoSpin®Gel and PCR clean up/NucleoBond® PC 100 kits were used, as well as Thermo Fisher Biosciences's GeneJET Plasmid Miniprep Kit and Agilent's QuikChange Site-Directed.

**Split YFP assay in *Xenopus* oocytes**. The first 425 aa of CaRhGC which covers the opsin and a part of the predicted coiled-coil sequence, was amplified by PCR with primer pair CaRhGC Kp5F (5′-cGGggtaccgataaggataacaatctccgtgga-3′) and CaRhGC 425Xh3R (5′-ccgCTCgaggatggcatcacagtt-3′), (#1, Supplementary Table 4). The KpnI and XhoI digested inserted to the pGEM-BiFC vector to generate a construct containing: 5′-YC (C-terminal 86 aa of YFP)-CaRh425aa-YN (N-terminal 155 aa of YFP)- 3′. After linearization (*Nhe*I), coding RNA (cRNA) was synthesized (AmpliCap-MaxT7 High Yield Message Maker Kit (Epicentre Biotechnologies)). 20 ng cRNA was injected in *Xenopus* oocytes, incubated in ND96 medium (96 mM NaCl, 2 mM KCl, 1 mM MgCl$_2$, 1 mM CaCl$_2$, 10 mM HEPES and 50 μg/ml gentamycin, pH 7.4) containing 1 μM all-trans retinal. Fluorescence pictures were taken 3 days after injection with confocal microscope (Leica DM6000).

**TEVC in oocytes**. Coding RNA (cRNA) for two electrode voltage clamp (TEVC) measurements was synthesized from linearized DNA (NheI) according to the manufacturer's instructions (mMESSAGE mMACHINE $^{TM}$ T7 kit, Invitrogen). cRNA of CaRhGC (2.5 ng), together with 5 ng of the cGMP-sensitive CNGA2 channel (rat olfactory neurons (gb: 6978671, NP_037060.1)) or 5 ng of the double mutated (C460W, E583M) version of the CNGA2 channel, sensitive to cAMP, was injected in *Xenopus laevis* oocytes. After incubation for 3–5 days in Ringer solution (96 mM NaCl, 5 mM KCl, 0.1 mM CaCl$_2$, 1 mM MgCl$_2$, 5 mM HEPES (pH 7.5)), supplemented with 1 μM all-trans retinal at 18 degrees, TEVC measurements were performed. A TURBO TEC-03 × amplifier (NPI Electronic), the pCLAMP 9.0 software (Molecular Devices), a XBO 75 W Xenon lamp (Osram) and a LS3 shutter (Vincent Associates UNIBLITZ) were used[13]. Microelectrodes were fabricated from borosilicate glass capillaries (1.50 mm outer diameter and 1.17 mm inner diameter) with a micropipette puller (model no. P-97, Sutter Instrument) and filled with 3 M KCl. Light was filtered with a 560-nm wideband filter (K55 Balzers, half-bandwidth, 60 nm) or a 530-nm filter (20BPF10-530 8C057, Newport; half-bandwidth, 9 nm). Light intensity was decreased with the help of neutral density filters. Oocytes were voltage-clamped at -40 mV and the composition of the extracellular buffer was 96 mM NaCl, 5 mM KCl, 0.1 mM CaCl$_2$, 1 mM MgCl$_2$, 5 mM HEPES (pH 7.5). Data were analyzed with Stimfit 0.13[43] and Clampfit 10.4 software (Molecular Devices LLC) as described before[13]. Figure 1c shows representative current traces for $n = 4$–9 cells co-expressing CaRhGC with the cGMP-sensitive CNGA2, for $n = 5$–11 cells expressing CaRhGC alone or with the cAMP-sensitive CNGA2. To determine the light intensity at half maximal response (EC$_{50}$ value), the slopes (20–80 %) of individual measurements were normalized to the maximum slope recorded in each oocyte. Normalized slopes were plotted against the light intensity and fitted exponentially using GraphPad Prism.

**cAMP and cGMP ELISA assay from whole oocyte lysates**. Oocytes injected with 30 ng cRNA of CaRhGC/BeRhGC/CaRhAC/BeRhAC variants were incubated at 18 °C for 3 days in Ringer solution supplemented with 1 μM all-trans retinal. Oocytes were either kept in the dark or illuminated for 1-4 min with green light (532 nm, 0.3 mW mm$^{-2}$). Five oocytes injected with the same construct were pooled and homogenized by pipetting in Sample Diluent (containing 0.1 N HCl and pH indicator, Arbor Assays). To remove cell debris samples were centrifuged at 12,000 r.p.m. for 6 min at room temperature. cAMP/cGMP within the oocyte lysate was quantified with the help of a DetectX High Sensitivity Direct Cyclic AMP/GMP Chemiluminescent Immunoassay Kit (Arbor Assays). Statistical significance between mean cAMP/cGMP concentrations light vs dark were tested with an unpaired two-sample *t*-test. Comparisons between different cyclases in the dark were performed using a one-way ANOVA followed by Tukey's multiple comparison tests.

**Xenopus oocyte membrane extraction and in vitro reaction**. Oocytes were frozen in liquid $N_2$ after expressing different constructs (30 ng cRNA, CaRhGCs, CaRhACs, BeRhACs) for 3–4 days. Homogenization buffer (83 mM NaCl, 2 mM $MgCl_2$, 1 × Protease Inhibitor Cocktail (Roche) and 10 mM HEPES, pH 7.5) were added in a ratio of 1 oocyte to 20 µl. Frozen oocytes were then homogenized on ice simply by pipetting with a 100 µl pipet. The yolk and cellular debris were sedimented by 500×g centrifugation at 4 °C for 20 min, and the supernatant was transferred to a new tube. The membrane fraction was then sedimented at 30,000 × g at 4 °C for 20 min. The membrane pellets were gently washed twice with 500 µl Homogenization buffer and re-suspended with Homogenization buffer at a ratio of 1 oocyte to 4 µl. A final 500×g centrifugation at 4 °C for 10 min was performed to remove the miss-transferred big cellular debris.

To start the enzymatic reaction, 2 µl of the membrane extract was mixed with 18 µl reaction buffer mix (for cAMP assay: 75 mM Tris-HCl, pH 7.4, 100 mM NaCl, 5 mM DTT, 5 mM $MgCl_2$, 1 mM ATP, 0.2 mM GTP) for cGMP assay (75 mM Tris-HCl, pH 7.4, 100 mM NaCl, 5 mM DTT, 10 mM $MgCl_2$, 2 mM GTP, 0.2 mM ATP) either in black tubes under infrared illumination (dark activity) or under illumination (532 nm, 0.3 mW $mm^{-2}$). To stop the reaction, 180 µl Sample Diluent (Chemiluminescent Immunoassay Kit) was added and cyclic nucleotides were quantified as described above. To calculate the enzymatic turnover, cNMP concentrations were determined at 3 different time points in the light (1, 4, and 7 min) and in the dark (5, 25, and 45 min). Turnover numbers were normalized to the amount of membranous protein, which was quantified through the YFP-tag emission (538 nm, Fluoroskan Ascent microplate fluorimeter) and compared to a YFP standard (Evrogen JSC; 1 mg $ml^{-1}$) of known concentration (dilution to 0, 5, 10, 20, 40, 80, and 160 ng with PBS). To determine membranous protein amount, Xenopus oocyte membranes were isolated as described above and the measured emission value of YFP-tagged rhodopsin cyclase was compared to the YFP standard curve. Immunoblot

One control oocyte or an oocyte expressing BeRhGC-YFP was homogenized by direct pipetting per 10 µl lysis buffer (0.1% Triton X-100, 4% SDS, 5% 2-mercaptoethanol, 10% glycerol, 0.002% bromophenol blue, 0.062 M Tris-HCl, pH 6.8). Samples were incubated for 1–2 h at room temperature and were either centrifuged for 5 min at 12000 g or not centrifuged before loading onto the running gel (NativePAGE™ Novex® 4–16% Bis-Tris Protein Gels, 1.0 mm, 10 wells, Protein ladder: (PageRuler Plus Prestained Protein Ladder #26619 (Thermo Scientific))). The proteins were then transferred to a Protran® Nitrocellulose Membrane for 100 min (0.8 mA $cm^{-2}$) by semi-dry transfer. The membrane was blocked with 1% BSA for 1 h at room temperature, washed with TBST twice for 10 min, incubated with GFP(FL)HRP sc-8334 (rabbit polyclonal IgG from Santa Cruz, 0.4 µg $ml^{-1}$) overnight at 4 °C, washed with TBST twice and once with TBS, and revealed in ECL solution (Thermo Scientific SuperSignal®West Pico Chemiluminescent Substrate). Images were obtained using a BIO-RAD ChemiDoc™ MP Imaging System at different exposure time.

**Purification of CaRh and full-length RhGC/RhAC**. The Catenaria rhodopsin domain (aa 1–396), the full-length CaRhGC and the double mutated CaRhAC were heterologously expressed (pFastBac) in Sf21 cells using the Bac-to-Bac Baculovirus Expression system (Thermo Fisher Scientific) according to the manufacturer's protocol. Sf21 cells were grown in Insect-XPRESS™-Media (LONZA) supplemented with 5 µM all-trans retinal (Sigma) during protein expression[44]. 72 h after virus infection cells were harvested, washed with buffer A (20 mM MOPS/HEPES pH 7.5, 100 mM NaCl, cOmplete Protease Inhibitor) and stored at −80 °C. All further purification steps were performed at 4 °C. Solubilization of the rhodopsin fragment was achieved in buffer A by adding n-dodecyl-β-D-maltoside (DDM) and cholesterol hemi-succinate to the thawed cells for final concentrations of 2% (w/v) and 0.5% (w/v), respectively. The full-length protein (CaRhGC/CaRhAC) was solubilized in a mixture of DDM/CHS/1,2-Dimirystoyl-sn-glycerol-3-phosphocholine (DMPC)/N,N-dimethyl-n-dodecylamine N-oxide (LDAO) in a final concentration of 2%/0.5%/0.01%/0.25% (w/v). After binding of protein to Ni-NTA resin (5 ml of HIS trap crude column, GE Healthcare) and washing the column with 10 column volumes of buffer A with 50 mM imidazole and 0.05%/0.01% DDM/CHS, protein was eluted with buffer A, 500 mM imidazole, 0.05%/0.01% DDM/CHS. Eluted protein was desalted (Hiprep 26/10 desalting column, GE Healthcare), pooled and loaded on a size exclusion column (HiLoad 16/600 Superdex 200 pg (GE Healthcare)). Purified protein was concentrated with an Amicon Ultra 100 kDa (Millipore) to an optical density of 1 at 540 nm. Concentration of purified protein was determined by absorption at 540 nm, $\varepsilon = 45,000$ $M^{-1}$ $cm^{-1}$.

**UV/Vis spectroscopy and laser flash photolysis**. Spectra from the Catenaria rhodopsin domain (1–396) were recorded in a Cary 50 Bio spectrophotometer (Varian, Inc.) at 20 °C at a spectral resolution of 1.6 nm. Light spectra were recorded after 30 s, 60 s, and 90 s of illumination with a green LED (530 nm, 0.54 mW $mm^{-2}$). Transient spectroscopy was performed on an LKS.60 flash photolysis system (Applied Photophysics Ltd.) at 22 °C. Excitation pulses of 10 ns (at 532 nm) were provided by a tunable Rainbow OPO/Nd:YAG laser system. Laser energy was adjusted to 15 mW/flash. Data analysis was performed with Matlab 7.01 software (The MathWorks). Singular value decomposition of representative data sets and data fitting by global analysis were performed with Glotaran 1.5.1 to identify

statistically significant components and time constants that were used in a sequential model for reconstruction of the 3D spectrum.

**Purification of the isolated cyclase domains**. His-tagged truncated cyclase domains (protein residues 443–626): CaGC, BeGC and CaAC (E497K/C566D) were expressed in E. coli C41 (DE3, Lucigen) cells at 37 °C in 6 × 800 ml culture volume of Luria-Bertani broth, containing 100 µg $ml^{-1}$ ampicillin. At $OD_{600} = 0.5$, cells were cooled to 18 °C for 1 h and expression was induced with 1 mM IPTG overnight at 18 °C. Cells were harvested and lysed (three rounds of French press). Cell debris were removed by two consecutive centrifugation steps: 10 min in 16,000×g, 4 °C and 1 h at 40,000×g, 4 °C and the supernatant was loaded on a Ni-NTA column (5 × 5 ml HisTrap HP, GE Healthcare). Following a washing step with 20 column volumes of 20 and 50 mM imidazole, the protein was eluted with 500 mM imidazole and desalted by a Hiprep 26/10 desalting column (GE Healthcare), equilibrated to 20 mM Tris/HCl, 50 mM NaCl, pH 8.0. Fractions of interest were pooled with a 50 kDa Amicon Ultra Filter (Millipore) and gelfiltrated (HiPrep 16/60 Sephacryl S-100 HR (GE Healthcare)). After elution with 20 mM Tris/HCl, 50 mM NaCl, pH 8.0, fractions containing the monomeric protein were pooled, concentrated with a 10 kDa Amicon Ultra Filter (Millipore) and concentration was measured (Nano Drop) with the cyclase molecular weight: 21.5 kDa. (CaGC extinction coefficient: 31002.00 l $mol^{-1}$ $cm^{-1}$, BeGC extinction coefficient: 31065.00 l $mol^{-1}$ $cm^{-1}$, CaAC extinction coefficient: 30940.00 l $mol^{-1}$ $cm^{-1}$).

**Enzymatic activity assays and pH dependence**. Activity assays (triplicates) at 22 °C were carried out in 50 mM HEPES, 100 mM NaCl, pH 7.5 (final volume 100 µl) with varying concentrations of NTP (GTP/ATP) and $Mn^{2+}$ (0.25–12 mM). The assay was started by addition of the respective purified enzyme (0.05 nmol full-length CaRhGC, 0.1 nmol BeRhGC, 2.3 nmol CaAC(E497K/C566D), 1.7 nmol CaGC, 1.3 nmol BeGC, 0.08 nmol CaRhAC(E497K/C566D). Four time points within the linear increase of cNMP were used to determine the initial velocity at a certain substrate concentration: for full-length CaRhGC (30 s, 60 s, 120 s, 180 s, 240 s), for truncated cyclases CaGC, BeGC, CaAC: 30 s, 1 min, 1.5 min, 2 min. The full-length protein was illuminated with green light (522 nm, 0.010 mW $mm^{-2}$, Adafruit NeoPixel NeoMatrix 8 × 8–64 RGB) during the incubation time, or kept in darkness. Enzymatic activity was stopped by flash freezing in liquid nitrogen and addition of 200 µl 0.1 N HCl. After centrifugation (90 s, 12,000×g, RT), the supernatant was filtered through a 0.2 µm chromafil filter (Macherey-Nagel) and 25 µl was applied on a C18 Reversed Phase High Pressure Liquid Chromatography (HPLC) column (SUPELCOSIL™ LC-18-T, 3 µm particle Size, 15 cm × 4.6 mm, Sigma Aldrich), equilibrated to 100 mM $K_2HPO_4$/$KH_2PO_4$, 4 mM tetra-butylammonium iodide, pH 5.9, 10 % methanol at a flow rate of 1.2 ml $min^{-1}$. Analyte elution was recorded via absorbance at 260 nm (retention time for cGMP ~7 min, for cAMP ~16 min). cNMP was quantified by peak analysis in Origin 8.5.5 (Originlab) and compared to peak values of cNMP standards (Sigma Aldrich) of known concentration. For each substrate concentration, cNMP concentrations were blotted against the time and the initial velocities were retrieved with the help of a linear fit. For the full-length constructs, the initial velocities of the dark samples were subtracted from the initial velocities of the illuminated samples. For Michaelis–Menten kinetics, initial velocities (full-length samples: corrected for dark activity) were plotted against the substrate concentration. $K_M$ and $v_{max}$ values could be assessed after applying a Hill fit ($y = v_{max} \times x^n/(k^n + x^n)$). $v_{max}$ values were normalized to the protein amount retrieved via Nanodrop for the isolated cyclases or spectroscopic absorption measurements at 540 nm for the full-length proteins.

To assess the substrate and enzymatic pH dependence, purified full-length CaRhGC/CaRhAC (E497K/C566D) or truncated cyclase CaGC/BeGC/CaAC (E497K/C566D) (protein amounts 11–55.36 µg) was incubated for 5 min with $Mn^{2+}$ (1 mM) and the adequate substrate GTP/ATP (1 mM) in a total volume of 100 µl of 50 mM buffer (MES for pH 5 and 6, HEPES for pH 7 and 7.5, Tris/Hcl for pH 8 and 9) and 100 mM NaCl. Assays were stopped and analyzed as stated before. Measurements were done in triplicate and the cNMP peak areas (mAU*min) per mg protein was determined. Data shown represent at least two independent repetitions.

To assess the inhibitory potential of ATP analogs, purified CaAC(E497K/C566D) (70 µg) was incubated with 1.4 mM ATP-Rp-αS/ATP-Sp-αS (Biolog) or APCPP (Jena Bioscience) in the presence of 0.5 mM ATP/Mn2 + (Sigma Aldrich) (100 mM NaCl, 50 mM HEPES, pH = 7.5) for 10 min. cAMP amounts were quantified as described before.

**Crystallization and structure solution of cyclases**. The cyclase domain CaAC (E497K/C566D) was co-crystallized with ATPαS (Jena Bioscience) at 20 °C. 0.45 µl of CaAC(E497K/C566D) solution (15.5 mg $ml^{-1}$ in 20 mM Tris/HCl pH 8, 50 mM NaCl, 7.5 mM $CaCl_2$, 7.5 mM $MgCl_2$, 10 mM ATPαS) was mixed with 0.45 µl reservoir solution (0.2 M potassium thiocyanate, 0.1 M sodium cacodylate pH 6.5, 25% w/v PEG 2000 MME) and equilibrated against 70 µl of reservoir solution (Clear strategy I, Molecular Dimensions). For data collection crystals were soaked in reservoir solution, supplemented with 7.5 mM $Ca^{2+}$, 7.5 mM $Mg^{2+}$ and 10 mM ATPαS and 20% glycerol (v/v), for cryo-protection. Diffraction data were collected at the Helmholtz-Zentrum Berlin für Materialien und Energie. (2016), the MX beamlines BL14.1-3 at BESSY II.[45] A full data set at 2.25 Å was retrieved and

processed with XDS[46], implemented in XDSAPP 2.0[47]. The structure was solved by molecular replacement using Phaser[48] with template coordinates derived from a high-resolution monomeric CaGC structure, which was solved with a loop depleted version of the mycobacterial AC (PDB: 4P2F). The CaAC model was improved by iterative manual fitting in COOT 0.8.7[49] and subsequent refinement cycles using BUSTER-TNT 2.10.2 -TNT 2.10.2 (BUSTER-TNT 2.X, Global Phasing Ltd, Sheraton House, Cambridge CB3 0AX, UK) and Phenix 1.10.1[50]. Figures were generated with Pymol (The PyMOL Molecular Graphics System,Version 1.8 Schrödinger, LLC.). The Stereo image is shown in Supplementary Fig. 15. The atomic coordinates and diffraction data are deposited at the Protein Data Bank, www.pdb.org: PDB ID code 5OYH.

**Hippocampal slice cultures and transfection.** Hippocampal slice cultures were prepared from P5-P7 Wistar rats (Janvier) and maintained in slice culture medium without antibiotics as described[51]. After 10 days to 3 weeks in vitro, hippocampal neurons were transfected by single-cell electroporation[52]. For the initial experiments with BeRhGC, CaRhGC, BeRhAC and CaRhAC, the person performing the electrophysiological recordings and analysis was blind to which plasmids were used for electroporation. In later experiments, the same individual performed all stages of the experiment and was no longer blinded.

**Hippocampal neuron electrophysiology.** Whole-cell patch clamp recordings were performed 6–15 days after electroporation using an Axopatch 200B amplifier. National Instruments A/D boards and Ephus software were used to record and control the experiment[53]. The extracellular recording solution contained: NaCl 119 mM, NaHCO$_3$ 26.2 mM, D-glucose 11 mM, KCl 2.5 mM, NaH$_2$PO$_4$ 1 mM, MgCl$_2$ 4 mM, CaCl$_2$ 4 mM, pH 7.4, 310 mOsm kg$^{-1}$, saturated with 95% O$_2$/5% CO$_2$. Recording temperature was 28–30 °C. The following were added to the perfusate to block synaptic activity unless otherwise indicated: NBQX 10 μM, CPPene 10 μM, picrotoxin 100 μM (Tocris). When light evoked currents were large enough to evoke action currents, the sodium channel blocker tetrodotoxin 1 μM was also added. Wash-in of the antagonists did not affect the light evoked currents. The intracellular solution contained: K-gluconate 135 mM, HEPES 10 mM, EGTA 0.2 mM, Na$_2$-ATP 4 mM, Na-GTP 0.4 mM, magnesium chloride (MgCl$_2$) 4 mM, ascorbate 3 mM, Na$_2$-phosphocreatine 10 mM, pH 7.2, 295 mOsm/kg. Patch electrodes were made from thick-walled borosilicate glass and had resistances of 3–5 MΩ. The liquid junction potential was measured (−14.1 to −14.4 mV) and compensated. Neurons were voltage-clamped at −70 mV. Series resistance was <25 MΩ (mean ± SEM = 10 ± 0.48 MΩ) and was not compensated during voltage clamp recordings. The bridge balance compensation circuitry was used during current clamp recordings. The sample was photo-stimulated through the objective (×40) using a four wavelength LED light source (Mightex Systems) coupled to the BX61WI microscope (Olympus) via the camera port through a multimode fiber (1.0 mm) and collimator (Thorlabs). The membrane response of the neurons was verified by somatic current injections (−400 pA to 400 pA).

The radiant power was measured with a silicon photodiode (Newport) in the specimen plane (Plan-Apochromat lens, 40 × 1.0 numerical aperture, Zeiss) and divided by the illuminated field (0.244 mm$^2$). When comparing the response to different wavelengths, the illumination intensity from the four-color LED was set to match as closely as possible, and the currents evoked were divided by the actual intensity of each wavelength used before normalization.

**Data analysis and statistics neuronal electrophysiology.** Analysis of the currents was performed with MATLAB, whereas graphs and curve-fitting were generated with GraphPad Prism 6.0. For the dose-response analysis, the slope was calculated between the 30 and 50% value of the peak response. Time to onset was defined as the intersection of the slope of the initial segment of the response and the start point of the illumination. The time of decay to half response was identified as the time after the stimulus when the response decayed to half of the value before the stimulus was off (sustained response). Error bars represent median and interquartile range it not otherwise stated.

**Data availability.** Data supporting the findings of this manuscript are available from the corresponding authors upon reasonable request. The sequence of BeRhGC is available under gb: AIC07007.1 (gb: KP731361 or gb: KF309499 (humanized codon-usage)), the sequence of CaRhGC is available under gb: MF939579. The atomic coordinates and diffraction data of the adenylyl cyclase in complex with ATP-Sp-αS are deposited at the Protein Data Bank, www.pdb.org: PDB ID code 5OYH. Plasmids will be available on Addgene.

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

## Acknowledgements

We wish to thank Maila Reh, Christina Schnick, Melanie Meiworm, Jan-Phillip Kehl, Iris Ohmert, Sabine Graf, and the Beamline staff for excellent technical assistance; Thomas Oertner for support and discussions; and Robert Lindner (Heidelberg) for constructive criticism on the manuscript. C.E.G. received funding from the Landesforschungsförderung Hamburg and the DFG (FOR2419, SPP1665). O.M.C. was supported by a DAAD scholarship. G.N. was supported by grants from the DFG (SFB 1047/A03, TRR 166/A03). G.N. and P.H. acknowledge support from the Louis-Jeantet-Foundation. P.H. received funding from the DFG (SFB1078) and from ERC (MERA). P. H. is Hertie Professor for neuroscience, supported by the Hertie Foundation.

## Author contributions

U.S.: photocurrents in oocytes, Elisas for BeRhGC/AC CaRhGC/AC mutations, protein expression, kinetic studies, crystallization of GC/ACs of Be and Ca, data collection at the synchrotron, analysis of the structure, and writing of the manuscript. M.B.: established crystallization, support of crystallization, data collection and resolving, and analysis of the structure, and contribution to writing of structural results and discussion. O.M.C.: photocurrents and 2-photon microscopy in neurons data collection and analysis, and participated in writing the mss. S.Y. and S.G.: cAMP and cGMP measurements of cyclase activity in whole oocytes and in oocyte membranes (in vitro assays), split GFP, mutations to AC and to reduce dark activity, kinetic studies. S.M.: support of kinetic studies. K.S.: protein expression and photocycle kinetics, and writing contribution to the spectroscopic part. G.N.: coordinator of mutagenesis and cAMP/cGMP measurements in oocyte membranes. C.E.G.: supervision of all neuronal experiments, project coordination with P.H., and substantial contribution to manuscript writing. P.H.: Project coordinator, supervision of protein expression, crystallization and biochemistry, writing of the manuscript with U.S. and C.E.G. with support from several others authors who read and approved the manuscript.

## Additional information

**Competing interests:** The authors declare no competing interests.

