## [Peer Review File · Nature Communications]

Reviewers' comments:

Reviewer #1 (Remarks to the Author):

This manuscript describes the use of a light-activated rhodopsin from *Catenaria anguillulae* (CaRhGC) to control cAMP or cGMP levels by light. The crystal structure of the AC domain (2.25 Å) gives detailed information about the nucleotide binding mode within this new class of enzyme rhodopsin. The rhodopsin domain from *Catenaria* proved to be more photostable than that from *Blastocladiella*, and the signaling state persisted longer, both of which might be highly desirable traits for use as an optogenetics tool. The paper is clear and the data are well described. The structures determined in this study will assist future efforts to create artificial light-regulated control modules as part of a general optogenetic toolkit.

I recommend publication provided that the following points can be addressed.

1. In the last part of the discussion, the activation of the new AC domain is compared to that of bPAC. The related OaPAC structure (also cited) has been solved to high resolution in the light and dark states, suggesting the allosteric mechanism involves a strong dynamic element. The authors of this paper apparently prefer the more rigid mechanical model suggested on the basis of bPAC, but perhaps half a sentence more could inform the reader that a role for flexibility is by no means ruled out.

2. Please indicate H-bonds in the legend of Figs. 7D, E, F - for example "black dashed lines indicate hydrogen bonds between 2.4 Å - 3.5 Å in length."

3. Show the Wilson B-factors and highest-resolution shell limits in Table 3.

Reviewer #2 (Remarks to the Author):

This manuscript presents an in-depth analysis of the *Catenaria anguillulae* rhodopsin-guanylyl cyclase, which is a light-activated enzyme producing guanylyl cyclase with a very large ratio of the activities in the light and in the dark. Moreover, they authors successfully mutated this enzyme into a light-activated adenylyl cyclase, though with a smaller light-to-dark ratio. The authors also present a crystal structure of the newly developed adenylyl cyclase domain in complex with a substrate analog.

The enzymological work is interesting, the crystallography is sound and the structure shows for the first time a nucleobase bound tightly in its binding pocket on such an enzyme. However, since this is the predicted binding mode, and the rest of the nucleotide clearly binds in a nonproductive manner, the structure does not add a great deal of novelty.

I therefore recommend not to publish the paper in Nature Communications but suggest the authors submit this manuscript to a more specialized journal.

In addition to this there are several issues that would need to be addressed as well. I will list them below:

*My main concern is with the proposed mechanism of activation. The authors argue that since the cyclase domain alone is active, whereas it is inactive in the dark when coupled to the rhodopsin domain, the rhodopsin acts as a clamp in the dark state, only to "release" the activity upon light activation of the rhodopsin. This may well be part of the story, but probably isn't all that is going on here. Looking at table 1, the light-activated rhodopsin-coupled cyclase is three times more active than the cyclase alone. A possible explanation for this would be that the rhodopsin in its light-activated state affects the dynamics of the cyclase. The authors should have addressed the

possibility of other models that are more complex than their current proposal.

*Indeed, the authors write in line 225 that they chose a particular dimer interface to analyse because it has the smallest overall B-factor. Does this mean that the B-factors of the others, and therefore the density of the other interfaces was too high for interpretation? That would be not be a problem in and of itself, but more likely a finding pointing to high structural dynamics in itself.

*Then there is the chemical mechanism. In line 251 the authors write about a "pseudobinuclear nucleophilic substitution SN2 mechanism"... This error was taken almost literally from the review cited. First of all binuclear is a term used for metal complexes. What is meant is probably "bimolecular". This, in turn, is wrong, too. There are first- and second order reactions, and under certain circumstances pseudo-first- and pseudo-second order reactions, but the order of the reaction is something other than its molecularity. Moreover SN2 already means "nucleophilic substitution", so the way this is phrased is double. But more errors are copied from the same review without critical evaluation:

l.252 "resulting in a negatively charged Pa during the transition state". In a pentacovalent transition state such as could be proposed for this reaction, indeed an extra negative charge develops on one of the oxygens of the alpha phosphate. But the way this is written implies that the phosphorous gets a negative charge, which is nonsense. Again, this wording was taken from the review cited, but is, depending on how it was meant, at best highly confusing, and at worst plain wrong.

Also (line 323): an in-line orientation of the reactants does not by itself mean that an associative mechanism, i.e. with a pentavalent phosphorous in the transition state, is formed. In fact, as the authors have little or no data that really touches upon the details of the transition state, such as a transition state analog or better kinetic isotope effects, I would not talk about it.

*Then, the way the paper is written seems to be aiming at an audience of experts in this field, as a lot of jargon is used. "light-to-dark ratio", for instance, and "after light-off". Such things should be explained. And in line 55: "the enzyme's activity declines with a delay of 300 ms.". Does it wait for 300 ms and then decay? Or does it decay with a 300 ms time constant? Please define such things clearly.

* Another case of a vague use of terms in the legends of fig 2 and 6: electroporation cannot be done with proteins. It was done with DNA coding for these proteins.

* l.267 how close is the ribose ring oxygen to S576? Why is this interaction called a "polar bond" (whatever that may be!), and not a hydrogen bond?

* fig.7. Although the 2Fo-Fc density (which is probably actually 2mFo-DFc) for the ligand is shown at a relatively high contour level (2 sigma) it would have been more convincing to use an omit map for the ligand, given that its B-factor is even higher than the average of the waters according to the crystallographic data table.

* Then there are some linguistic/typographic problems:

l.51. "putative coiled coil linker" – is the linker putative? Or it's coiled coil structure?

l. 74 . "extra helices"? on top of what? the rhodopsin? that becomes clear only afterwards

l.71. 2,6 x higher should be 2.6x higher

l.350 "Hereby"? Whereby? The authors mean that the role of the sulfur in determining ligand conformation is unclear.

l.393 N-terminal YFP tagged  N-terminally YFP-tagged (adjective, not adverb!)

Table 1. I am pretty sure the Km in kcat/Km should not be in subscript...

*As a final note, the structure report from the PDB asserts that the R-epimer of the ligand has been built in. This is a mistake by the PDB, the authors have correctly built in the S-epimer, as

shown in their figures. There is probably something wrong with the ligand database; if I were the authors I would ask the PDB to change this, otherwise people who download the structure might be confused.

Reviewer #3 (Remarks to the Author):

Scheib and coauthors described new optogenetic tools, green-light activated rhodopsin-guanylyl cyclase (GC) and adenylyl cyclase (AC). Because of the breadth of cGMP and cAMP signaling in animals, improved optogenetic tools are undoubtedly welcome. Specifically, (i) the researchers identified and characterized the rhodopsin GC from fungus *Catenaria* (CaRhGC) -- in vitro, in oocytes and in neuronal culture. They showed that CaRhGC is more stable in high light than the related rhodopsin GC from fungus *Blastocladiella* (BeRhGC = CyclOp). (ii) They converted the GC into the photoactivated adenylyl cyclase (AC) by mutagenizing residues in the substrate binding pocket. (iii) They also determined the X-ray structure of the catalytic domain of the designed AC. Each of these accomplishments is a step forward, but an incremental step. Together, these advances amount to a good, solid paper, yet it is unclear whether they amount to a high-profile paper.

(i) When expressed in neurons, currents evoked by brief irradiation of neurons expressing CaRhGC were similar to the currents in neurons expressing BeRhGC (Fig. 2c-2e). There were some differences in kinetics. Were these differences significant to suggest that CaRhGC outperforms BeRhGC? Disturbingly (page 5, lines 106-8), photocurrents were evoked in ~70% of neurons expressing BeRhGC but only in ~30% of neurons expressing CaRhGC. Is retinal incorporation the same in both enzymes? Isn't this a major problem for CaRhGC that outweighs its photostability and somewhat higher activity advantage?

(ii) The GC-to-AC conversion by mutagenesis of substrate-binding residues has been known for two decades (as pointed out in the paper), so the conversion of CaRhGC to CaRhAC in itself is not particularly impressive. CaRhGC has a somewhat higher dynamic range (Fig. 5b) compared to the similarly designed BeRhAC (reference 14), which is an improvement. However, it is unclear how it compares to the blue-light ACs referenced in the manuscript or to the OptoXR and its improved variants (not referenced here).

(iii) The 2.25Å-crystal structure of the catalytic dimer of the engineered AC domains is a welcome addition to previously determined structures of type III nucleotidyl cyclases. It is unclear that it adds much to the understanding of substrate specificity of ACs and GCs or catalytic mechanism. The lengthy description of the structure seems to be confirmatory of the conclusions reached from the AC structures described previously. The structure of a homodimeric GC (from *Chlamydomonas*) is unfortunately not even cited. Most disappointing is that no new insights emerged into the photoactivation mechanism.

Reviewer #4 (Remarks to the Author):

This is a very informative, well-designed and well-executed study which characterized the properties of rhodopsin-guanylyl cyclase (RhoGC) from *Catenaria anguilulae* (CaCylcOp). Earlier studies had previously explored the properties of rhodopsin guanylyl cyclase from *Blastocladiella emersonii* (BeCylcOp) which has a 77% sequence homolog with CaCylcOp.

Importantly, the CaAC structure elucidation provided a basis and confirmation for much earlier

work to understanding how the E497K and C566D mutations resulted in the switch in specificity from GC to AC. It also revealed difference compared to other adenylyl cyclase structures. Overall, the work provides evidence that the rhodopsin structure acts to "clamp" the CaGC domain (or CaAC domain) that is relieved during the photocycle, although it does not reveal the molecular mechanism of the clamp.

The paper should be published after the following revisions:

1) Several experiments indicate that CaCylcOp has superior properties as an optogenetic tool compared to BeCyclOP. For example, the onset of photocurrent measured in neurons was much shorter for CaRhGC compared to BeRhGC (23 ms vs. 120 ms) allowing faster response times when used as a light trigger for GMP production. However, even though the photocurrents for CaRhGC and BeRhGC were similar, it was also found that CaRhoGC photocurrents were evoked in only 30% of neurons compared to 70% for BeRhGC. No explanation was offered for this observation. It would therefore be very helpful for the reader if possible causes were listed and how experiments might be designed to test these possibilities.

2) The photocycle characterization of the rhodopsin domain revealed that CaRhGC has a very slow rise time (31 ms) for the M intermediate and even longer than previously reported time for BeRhGC (8 ms). This is an unusual feature for a microbial rhodopsin (e.g. bacteriorhodopsin is only 40 microseconds) and would be worth some discussion.

3) While M is proposed to be the active state for GC activity, there is no direct evidence to support this claim. Furthermore, no data is presented to support the claim that CaRhGC has a higher photo-stability compared to BeRhGC. Finally, it should be noted that these measurements are made in detergent and not a lipid bilayer membrane where the kinetics can be considerably different.

4) Table I compares the enzymatic parameters for full-length CaRhGC and truncated GC which in some cases causes dramatic change such as for v_{max} and k_{cat} . While the BeGC (truncated) is also listed, the full-length BeRhGC is not. This would be very useful, especially since an important theme of the paper is to compare the properties of CaRhGC and BeRhGC.

Reviewers' comments:

Reviewer #1 (Remarks to the Author):

*This manuscript describes the use of a light-activated rhodopsin from *Catenaria anguillulae* (CaRhGC) to control cAMP or cGMP levels by light. The crystal structure of the AC domain (2.25 Å) gives detailed information about the nucleotide binding mode within this new class of enzyme rhodopsin. The rhodopsin domain from *Catenaria* proved to be more photostable than that from *Blastocladiella*, and the signaling state persisted longer, both of which might be highly desirable traits for use as an optogenetics tool. The paper is clear and the data are well described. The structures determined in this study will assist future efforts to create artificial light-regulated control modules as part of a general optogenetic toolkit.*

I recommend publication provided that the following points can be addressed.

1. In the last part of the discussion, the activation of the new AC domain is compared to that of bPAC. The related OaPAC structure (also cited) has been solved to high resolution in the light and dark states, suggesting the allosteric mechanism involves a strong dynamic element. The authors of this paper apparently prefer the more rigid mechanical model suggested on the basis of bPAC, but perhaps half a sentence more could inform the reader that a role for flexibility is by no means ruled out.

This is a good point, indeed there may well be a dynamic element in the RhACs (and GCs). We now include some discussion of the OaPAC activation mechanism and make clear that we do not rule out a role for flexibility (see line 379-381).

2. Please indicate H-bonds in the legend of Figs. 7D, E, F - for example "black dashed lines indicate hydrogen bonds between 2.4 Å - 3.5 Å in length."

H-bonds are now indicated

3. Show the Wilson B-factors and highest-resolution shell limits in Table 3.

We now show the Wilson B-factors and the limits of the highest-resolution shell (2.33 - 2.249 Å) in table 3.

Reviewer #2 (Remarks to the Author):

*This manuscript presents an in-depth analysis of the *Catenaria anguillulae* rhodopsin-guanylyl cyclase, which is a light-activated enzyme producing guanylyl cyclase with a very large ratio of the activities in the light and in the dark. Moreover, they authors successfully mutated this enzyme into a light-activated adenylyl cyclase, though with a smaller light-to-dark ratio. The authors also present a crystal structure of the newly developed adenylyl cyclase domain in complex with a substrate analog.*

The enzymological work is interesting, the crystallography is sound and the structure shows for the first time a nucleobase bound tightly in its binding pocket on such an enzyme. However, since this is the predicted binding mode, and the rest of the nucleotide clearly binds in a nonproductive manner, the structure does not add a great deal of novelty.

I therefore recommend not to publish the paper in Nature Communications but suggest the authors submit this manuscript to a more specialized journal.

In addition to this there are several issues that would need to be addressed as well. I will list them below:

1. My main concern is with the proposed mechanism of activation. The authors argue that since the cyclase domain alone is active, whereas it is inactive in the dark when coupled to the rhodopsin domain, the rhodopsin acts as a clamp in the dark state, only to "release" the activity upon light activation of the rhodopsin. This may well be part of the story, but probably isn't all that is going on here. Looking at table 1, the light-activated rhodopsin-coupled cyclase is three times more active than the cyclase alone. A possible explanation for this would be that the rhodopsin in its light-activated state affects the dynamics of

the cyclase. The authors should have addressed the possibility of other models that are more complex than their current proposal.

We fully agree with the reviewer that in addition to acting as an inactivation clamp in darkness, illumination of the full-length Ca/BeRhGC further increases v_{\max} compared to the isolated cyclase domains. We modified the discussion to make this clear and we extended the activation mechanism model, see line 289 ff.

2. Indeed, the authors write in line 225 that they chose a particular dimer interface to analyse because it has the smallest overall B-factor. Does this mean that the B-factors of the others, and therefore the density of the other interfaces was too high for interpretation? That would be not be a problem in and of itself, but more likely a finding pointing to high structural dynamics in itself.

As stated in line 217 we obtained well interpretable electron density for all 8 dimers and dimer interfaces with the average B-factor for the ligand ranging from $\sim 41 \text{ \AA}^2$ (chain A) to $\sim 76 \text{ \AA}^2$ (chain J). In all 8 dimer interfaces the electron density shows the same conformation of the ligand and its protein surrounding (except for the $\beta 4/5$ loop). Some particular residues e.g. the sidechain of R577 or some water molecules are not equally well resolved for all dimers, therefore we specify the dimer composed of chain A/B as the basis of our description. These differences most likely result from the overall packing of the dimer within the unit cell.

3. Then there is the chemical mechanism. In line 251 the authors write about a "pseudobinuclear nucleophilic substitution SN2 mechanism"... This error was taken almost literally from the review cited. First of all binuclear is a term used for metal complexes. What is meant is probably "bimolecular". This, in turn, is wrong, too. There are first- and second order reactions, and under certain circumstances pseudo-first- and pseudo-second order reactions, but the order of the reaction is something other than its molecularity. Moreover SN2 already means "nucleophilic substitution", so the way this is phrased is double.

We have removed the term "pseudobinuclear" which we originally understood to refer to a reaction catalyzed by two metal centers (ion A and B), albeit only one (ion A) being actively involved (therefore "pseudo"). We re-worded the sentence to "ATP cleavage and cyclization is considered to follow an intramolecular nucleophilic substitution (S_N2), which is initiated through the attack of ribose 3'-OH oxygen at P_α , resulting in a negatively charged α -phosphate during the transition state.", line 244.

But more errors are copied from the same review without critical evaluation: I.252 "resulting in a negatively charged P_α during the transition state". In a pentacovalent transition state such as could be proposed for this reaction, indeed an extra negative charge develops on one of the oxygens of the alpha phosphate. But the way this is written implies that the phosphorous gets a negative charge, which is nonsense. Again, this wording was taken from the review cited, but is, depending on how it was meant, at best highly confusing, and at worst plain wrong.

Since P_α was misleading we replaced it by α -phosphate line 244, 276, 345.

Also (line 323): an in-line orientation of the reactants does not by itself mean that an associative mechanism, i.e. with a pentavalent phosphorous in the transition state, is formed. In fact, as the authors have little or no data that really touches upon the details of the transition state, such as a transition state analog or better kinetic isotope effects, I would not talk about it.

The present study does not focus on the elucidation of the mechanistic aspects of the enzymatic reaction. The described mechanism including the proposed transition state is not derived from our data, instead it is mainly based on the stereochemistry previously observed for the reaction of adenylyl cyclases in general that shows inversion of the configuration at P_α (Gerlt et al., JBC (1980): 255, pp.331-334; now included as reference in the text, line 244). The overview about the current understanding of the mechanism is intended to facilitate the discussion regarding the nucleotide conformation in CaAC, the position of crucial

residues defining the ATP vs. GTP selectivity (e.g. R577), the overall conformation of the dimer and the possible mechanism of photoregulation.

**Then, the way the paper is written seems to be aiming at an audience of experts in this field, as a lot of jargon is used. "light-to-dark ratio", for instance, and "after light-off". Such things should be explained. And in line 55: "the enzyme's activity declines with a delay of 300 ms.". Does it wait for 300 ms and then decay? Or does it decay with a 300 ms time constant? Please define such things clearly.*

Light-to-dark ratio refers to the ratio of the activity in light to the activity in darkness, we hope this is clearer when we write "light/dark activity ratio" (l. 23, 178) and we replaced "L/D" by light/dark in l. 182. *Light-off* means: after the light has been switches off, which should be clear.

4. Another case of a vague use of terms in the legends of fig 2 and 6: electroporation cannot be done with proteins. It was done with DNA coding for these proteins.

This has been corrected.

5. 1.267 how close is the ribose ring oxygen to S576? Why is this interaction called a "polar bond" (whatever that may be!), and not a hydrogen bond?

We changed this to hydrogen bond, line 261.

6. fig. 7. Although the 2Fo-Fc density (which is probably actually 2mFo-DFc) for the ligand is shown at a relatively high contour level (2 sigma) it would have been more convincing to use an omit map for the ligand, given that its B-factor is even higher than the average of the waters according to the crystallographic data table.

Figure 7 now includes an mFo-DFc omit map.

7. Then there are some linguistic/typographic problems:

I.51. "putative coiled coil linker" – is the linker putative? Or it's coiled coil structure?

Clarified – the *structure* is putative.

I. 74 . "extra helices"? on top of what? the rhodopsin? that becomes clear only afterwards

Re-phrased

I.172. 2,6 x higher should be 2.6x higher

Good eye, thanks.

I.350 "Hereby"? Whereby? The authors mean that the role of the sulfur in determining ligand conformation is unclear.

Clarified

I.393 N-terminal YFP tagged  N-terminally YFP-tagged (adjective, not adverb!)

Corrected.

Table 1. I am pretty sure the Km in kcat/Km should not be in subscript...

Corrected

8. As a final note, the structure report from the PDB asserts that the R-epimer of the ligand has been built in. This is a mistake by the PDB, the authors have correctly built in the S-epimer, as shown in their figures. There is probably something wrong with the ligand database; if I were the authors I would ask the PDB to change this, otherwise people who download the structure might be confused.

Thank you for having noticed this and pointing it out. We have been in contact with the PDB to change the ligand ID to the S-epimer.

Reviewer #3 (Remarks to the Author):

Scheib and coauthors described new optogenetic tools, green-light activated rhodopsin-guanylyl cyclase (GC) and adenylyl cyclase (AC). Because of the breadth of cGMP and cAMP signaling in animals, improved optogenetic tools are

undoubtedly welcome. Specifically, (i) the researchers identified and characterized the rhodopsin GC from fungus Catenaria (CaRhGC) -- in vitro, in oocytes and in neuronal culture. They showed that CaRhGC is more stable in high light than the related rhodopsin GC from fungus Blastocladiella (BeRhGC = CyclOp). (ii) They converted the GC into the photoactivated adenylyl cyclase (AC) by mutagenizing residues in the substrate binding pocket. (iii) They also determined the X-ray structure of the catalytic domain of the designed AC. Each of these accomplishments is a step forward, but an incremental step. Together, these advances amount to a good, solid paper, yet it is unclear whether they amount to a high-profile paper.

(i) When expressed in neurons, currents evoked by brief irradiation of neurons expressing CaRhGC were similar to the currents in neurons expressing BeRhGC (Fig. 2c-2e). There were some differences in kinetics. Were these differences significant to suggest that CaRhGC outperforms BeRhGC? Disturbingly (page 5, lines 106-8), photocurrents were evoked in ~70% of neurons expressing BeRhGC but only in ~30% of neurons expressing CaRhGC. Is retinal incorporation the same in both enzymes? Isn't this a major problem for CaRhGC that outweighs its photostability and somewhat higher activity advantage?

We were also disturbed by this observation and therefore had included the 'success' data in the text and supplementary table 1. We have now produced and tested 3 new constructs and found that adding a YFP tag to the N-terminus greatly improved the reliability without changing the other properties. We additionally found that addition of a mycHis tag to the C-terminus of YFP-CaRhGC (l. 185, 321 - 323) worsened it but that changing from a human or mouse codon optimization strategy made almost no difference. This is now mentioned in the results section, the overview of all results in Supplementary Table 1. Figure 2 now features YFP-CaRhGC, the CaRhGC remains for comparing with the equivalent BeRhGC. l. 88 - 97.

(ii) The GC-to-AC conversion by mutagenesis of substrate-binding residues has been known for two decades (as pointed out in the paper), so the conversion of CaRhGC to CaRhAC in itself is not particularly impressive. CaRhGC has a

somewhat higher dynamic range (Fig. 5b) compared to the similarly designed BeRhAC (reference 14), which is an improvement. However, it is unclear how it compares to the blue-light ACs referenced in the manuscript or to the OptoXR and its improved variants (not referenced here).

We agree completely with the reviewer that the mutations to convert the GC to an AC does not constitute a breakthrough. We do however stand behind the claim that CaRhAC is truly superior to BeRhAC as an optogenetic tool.

In the discussion we now clearly state how we see CaRhAC in comparison to bPAC, the optoXRs and the jellyfish opsins. CaRhAC raises cAMP much faster than bPAC. If a LOT of cAMP is desired than bPAC is still the tool of choice. The OptoXRs and jellyfish opsins require two extra components in the cell of interest to work and the G proteins may well activate other intracellular signaling pathways in addition to raising cAMP – there is no question that they are also important tools but they don't only raise cAMP. The added discussion can be found in **lines 328-333**.

*(iii) The 2.25Å-crystal structure of the catalytic dimer of the engineered AC domains is a welcome addition to previously determined structures of type III nucleotidyl cyclases. It is unclear that it adds much to the understanding of substrate specificity of ACs and GCs or catalytic mechanism. The lengthy description of the structure seems to be confirmatory of the conclusions reached from the AC structures described previously. The structure of a homodimeric GC (from *Chlamydomonas*) is unfortunately not even cited. Most disappointing is that no new insights emerged into the photoactivation mechanism.*

To our knowledge the structure of CaAC demonstrates the nucleotide binding of RhGCs for the first time on a structural level. Recently Kumar et al. (Kumar et al., 2017) showed the isolated guanylyl/adenylyl cyclase of *Blastocladiella emersonii* as monomer or unusual head-to-head dimer (without a bound ligand). This posed the question whether cyclases of the new enzyme-rhodopsin class adopt an alternative reaction mechanism. In contrast, the CaAC structure showed an antiparallel homodimeric arrangement as seen for other type III cyclases and suggests that this class of cyclase indeed uses a similar catalytic

pathway. We strengthened this point in line 205-207, 334-337 and added the citation of other GC structures, line 211.

The observed β 4/5 flexibility of CaAC supports some conclusions regarding the intramolecular signal propagation. We of course hoped for and invested quite some efforts trying to crystalize the full-length protein but perhaps not surprisingly for such a large 8 transmembrane spanning protein we did not yet succeed. The AC structure serves, however, as an important starting point towards understanding the signal transduction within the full-length protein. In particular, sophisticated biophysical methods like electron microscopy and advanced spectroscopy as FTIR and EPR will benefit from the AC-structure for further interpretation. Moreover, our structure provides the basis for rational guided mutagenesis to gain detailed functional understanding and to generate variants with differing kinetics and/or specificity.

Reviewer #4 (Remarks to the Author):

*This is a very informative, well-designed and well-executed study which characterized the properties of rhodopsin-guanylylcyclase (RhoGC) from *Catenaria anguilulae*(CaCylcIOP). Earlier studies had previously explored the properties of rhodopsin guanylylcylase from *Blastoclaadiella emersonii* (BeCyclOP) which has a 77% sequence homolog with CaCylOP.*

Importantly, the CaAC structure elucidation provided a basis and confirmation for much earlier work to understanding how the E497K and C566D mutations resulted in the switch in specificity from GC to AC. It also revealed difference compared to other adenylyl cyclase structures. Overall, the work provides evidence that the rhodopsin structure acts to "clamp" the CaGC domain (or CaAC domain) that is relieved during the photocycle, although it does not reveal the molecular mechanism of the clamp.

The paper should be published after the following revisions:

1) Several experiments indicate that CaCylcIOP has superior properties as an optogenetic tool compared to BeCyclOP. For example, the onset of photocurrent measured in neurons was much shorter for CaRhGC compared to BeRhGC (23

ms vs. 120 ms) allowing faster response times when used as a light trigger for GMP production. However, even though the photocurrents for CaRhGC and BeRhGC were similar, it was also found that CaRhGC photocurrents were evoked in only 30% of neurons compared to 70% for BeRhGC. No explanation was offered for this observation. It would therefore be very helpful for the reader if possible causes were listed and how experiments might be designed to test these possibilities.

To answer this concern we have now cloned and tested 3 additional versions of CaRhGC and 'cured' the unreliability. The 'cure' was achieved by adding the N-terminal YFP-tag and ensuring that there was no mycHis tag on the C-terminus. We also had both humanized and murine-optimized codons in our constructs and have directly tested this and could conclude that this made little difference. Figure 2 has been changed to feature YFP-CaRhGC, CaRhGC is left for the direct comparison in neurons to BeRhGC and additional data is included in Supplementary Table 1 and supplementary Figures 3. And the relevant sections of text have been modified lines 88-97.

2) The photocycle characterization of the rhodopsin domain revealed that CaRhGC has a very slow rise time (31 ms) for the M intermediate and even longer than previously reported time for BeRhGC (8 ms). This is an unusual feature for a microbial rhodopsin (e.g. bacteriorhodopsin is only 40 microseconds) and would be worth some discussion.

The M rise time of CaRhGC is indeed rather slow. There are other examples of microbial rhodopsins with similarly slow formation of the M intermediate e.g. histidine kinase rhodopsins (HKR1) that we mention in line 298-300.

3) While M is proposed to be the active state for GC activity, there is no direct evidence to support this claim. Furthermore, no data is presented to support the claim that CaRhGC has a higher photo-stability compared to BeRhGC. Finally, it should be noted that these measurements are made in detergent and not a lipid bilayer membrane where the kinetics can be considerably different.

To determine whether the M state is the actual signaling state is generally challenging, especially in this case as the current readout in the electrophysiology experiments is indirect. That being said, in the neuron electrophysiology experiments, the onset of the photocurrents was around 20 ms meaning the signaling state must have been reached by this time. This time corresponds well to the formation of the M state (remembering these recordings were at 31 °C not RT). Alternatively an earlier state could be the signaling state. We now discuss this point and give an explanation why we think that the M state can be correlated with the signaling state, line 300-305. The higher photostability of CaRh is presented in Fig.3a. High intensity illumination for 90 seconds kept the spectrum unchanged (in contrast to BeChGC, which substantially bleaches under these conditions (Scheib et al. 2015, Penzkofer et al. 2017, cited in line 113). But we agree that the stability could be different in cells, where it is more difficult to quantify. However, based on the data we are convinced that the relative photostability is increased for CaRhGC.

4) Table 1 compares the enzymatic parameters for full-length CaRhGC and truncated GC which in some cases causes dramatic change such as for v_{max} and k_{cat} . While the BeGC (truncated) is also listed, the full-length BeRhGC is not. This would be very useful, especially since an important theme of the paper is to compare the properties of CaRhGC and BeRhGC.

We agree that these data were missing. For resubmission of the manuscript we re-cloned BeRhGC into an appropriate vector for insect cell expression and purified it as a detergent-solubilized protein. We found v_{max} of illuminated BeRhGC was ~6x less than for CaRhGC and that BeRhGC had a reduced light-to-dark activity ratio. We included the data in Supplementary Fig. 4, and Table 1 and described our results in line 136-141. We further discussed these data in lines 309.

REVIEWERS' COMMENTS:

Reviewer #3 (Remarks to the Author):

My concerns have been adequately addressed in the revised manuscript.

Reviewer #4 (Remarks to the Author):

The authors' revisions have satisfactorily addressed all of the comments/suggestions in my review of the original manuscript. I recommend that the paper now be published without further revisions.